# Novel Object Synthesis via Adaptive Text-Image Harmony

**Zeren Xiong[1], Zedong Zhang[1], Zikun Chen[1], Shuo Chen[2],**
**Xiang Li[3], Gan Sun[4], Jian Yang[1], Jun Li[1]***
[1]School of Computer Science and Engineering,
Nanjing University of Science and Technology, Nanjing, 210094, China
[2]RIKEN,   [3]College of Computer Science, Nankai University, Tianjing, 300350, China
[4]College of Automation Science and Engineering,
South China University of Technology, Guangzhou, 510640, China
{zandyz,csjyang,junli}@njust.edu.cn,   {xzr3312,zikunchencs,sungan1412}@gmail.com
xiang.li.implus@nankai.edu.cn,   shuo.chen.ya@riken.jp

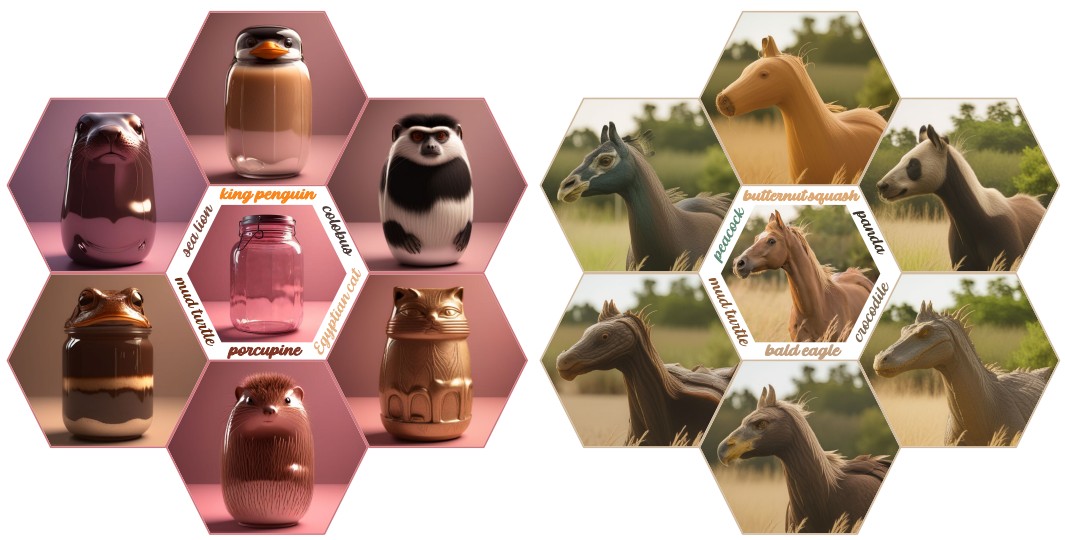

Figure 1: We propose a straightforward yet powerful approach to generate combinational objects from a given object text-image pair for novel object synthesis. Our algorithm produces these combined object images using the central image and its surrounding text inputs, such as *glass jar* (image) and *porcupine* (text) in the left picture, and *horse* (image) and *bald eagle* (text) in the right picture.

## Abstract

In this paper, we study an object synthesis task that combines an object text with an object image to create a new object image. However, most diffusion models struggle with this task, *i.e.*, often generating an object that predominantly reflects either the text or the image due to an imbalance between their inputs. To address this issue, we propose a simple yet effective method called Adaptive Text-Image Harmony (ATIH) to generate novel and surprising objects. First, we introduce a scale factor and an injection step to balance text and image features in cross-attention and to preserve image information in self-attention during the text-image inversion diffusion process, respectively. Second, to better integrate object text and image, we design a balanced loss function with a noise parameter, ensuring both

---

*Corresponding author

38th Conference on Neural Information Processing Systems (NeurIPS 2024).

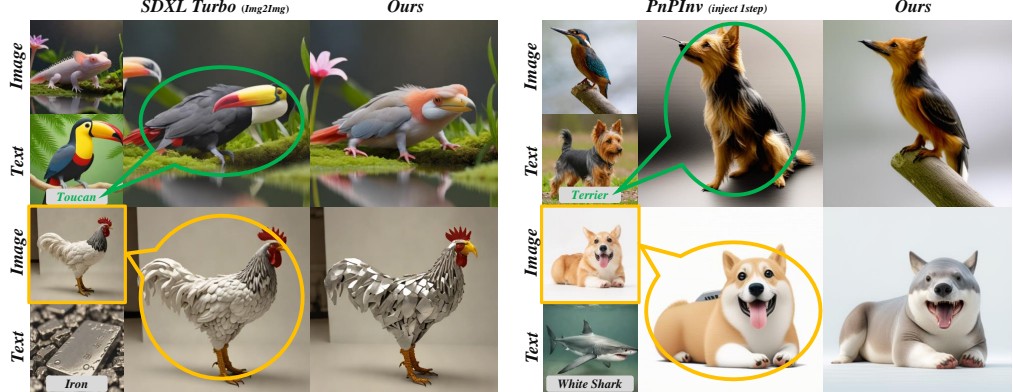

Figure 2: **Imbalances between text and image in diffusion models.** Using SDXL-Turbo [56] (left) and PnPinv [27] (right), the top pictures show a tendency for generated objects to align with textual content (green circles), while the bottom pictures tend to align with visual aspects (orange circles). In contrast, our approach achieves a more harmonious integration of both object text and image.

optimal editability and fidelity of the object image. Third, to adaptively adjust these parameters, we present a novel similarity score function that not only maximizes the similarities between the generated object image and the input text/image but also balances these similarities to harmonize text and image integration. Extensive experiments demonstrate the effectiveness of our approach, showcasing remarkable object creations such as *colobus-glass jar* in Fig. 1. Project Page.

# 1   Introduction

Image synthesis from text or/and image using diffusion models such as Stable Diffusion [51], SDXL [56], and DALL·E3 [43] has gained considerable attention due to their impressive generative capabilities and practical applications, including editing [6; 75] and inversion [24; 61]. Many of these methods focus on object-centric diffusion, utilizing textual descriptions to manipulate objects within images through operations like composition [58], addition [36; 15], removal [63], replacement [7], movement [29], and adjustments in size, shape, action, and pose [17]. In contrast, we study an object synthesis task that creates a new object image by combining an object text with an object image. For instance, combining *kingfisher* (image) and *terrier* (text) results in a new and harmonious terrier-like kingfisher object, as shown in the right-side of Fig. 2.

To implement object text-image fusion, most diffusion models, such as SDXL-Turbo [56], often use cross-attention [24] to integrate the input text and image. However, the cross-attention frequently results in imbalanced outcomes, as evidenced by the following observations. On the left side of Fig. 2, when inputting an *axolotl* (image) and a *toucan* (text), SDXL-Turbo only generates an image of a toucan, showing a bias towards the toucan text (green circles). Conversely, when inputting a *rooster* (image) and an *iron* (text), it produces an image of a rooster, which closely resembles the original rooster image (orange circles). These observations reveal that the text (or image) feature often suppresses the influence of the image (or text) feature during the diffusion process, leading to a failed fusion. To mitigate the image degeneration, Plug-and-Play [61] can inject the guidance image features into self-attention. Unfortunately, even with the application of the best inversion editing method, PnPinv [27], which incorporates the plug-and-play inversion into diffusion-based editing methods for improved performance, we still observe similar imbalances, as shown on the right-side of Fig. 2. This arises an important problem: *how can we balance object text and image integration?*

To address this problem, we propose an **A**daptive **T**ext-**I**mage **H**armony (**ATIH**) method for novel object synthesis, as shown in Fig. 3. First, during the inversion diffusion process, we introduce a scale factor $\alpha$ to balance text and image features in cross-attention, and an injection step $i$ to preserve image information in self-attention for adaptive adjustment. Second, the inverted noise maps adhere to the statistical properties of uncorrelated Gaussian white noise, which increases editability [46]. However, they are preferable for approximating the feed-forward noise maps, thereby enhancing fidelity. To better integrate object text and image, we treat sampling noise as a parameter in designing

a balanced loss function, which strikes a balance between reconstruction and Gaussian white noise approximation, ensuring both optimal editability and fidelity of the object image. Third, we present a novel similarity loss that considers both $i$ and $\alpha$. This loss function not only maximizes the similarities between the generated object image and the input text/image but also balances these two similarities to harmonize text and image integration. Furthermore, we employ the *Golden Section Search* [47] algorithm to quickly find the optimal parameters $\alpha$ and $i$. Therefore, our ATIH method is capable of generating novel object combinations. For instance, an **iron-like rooster** is produced by merging the image *rooster* with the text *iron*, resulting in a rooster image with an iron texture, as shown in Fig. 2.

Overall, our contributions can be summarized as follows: **(1)** To the best of our knowledge, we are the first to propose an adaptive text-image harmony method for generating novel object synthesis. The key idea is to achieve a balanced blend of object text and image by adaptively adjusting a scale factor and an injection step in the inversion diffusion process, ensuring their effective harmony. **(2)** We introduce a novel similarity score function that incorporates the scale factor and injection step. This aims to balance and maximize the similarities between the generated image and the input text/image, achieving a harmonious integration of text and image. **(3)** Experimental results on PIE-bench [26] and ImageNet [53] demonstrate the effectiveness of our method. Our approach shows superior performance in creative object combination compared to state-of-the-art image-editing and creative mixing methods. Examples of these creative objects, such as *sea lion-glass jar*, *African chameleon-bird*, and *corgi-cock* are shown in Figs. 1, 6, and 8.

## 2 Related Work

**Text-to-Image Generation** The rapid development of generative models based on diffusion processes has advanced the state-of-the-art for tasks [12; 21; 33] like text-to-image synthesis [22; 31], image editing [64; 2], and style transfer [65; 23; 35]. Large-scale models such as Stable Diffusion [51], Imagen [55], and DALL-E [49] have demonstrated remarkable capabilities. Sdxlturbo [56] introduced a distillation method that further enhances efficiency by reducing the steps needed for high-quality image generation. Our method utilizes Sdxlturbo for adaptive and innovative object fusion, preserving the original image's layout and details while requiring only the textual description of the target object.

**Text Guided Image Editing.** Diffusion models have garnered significant attention for their success in text-to-image generation and text-driven image editing using natural language descriptions. Early studies [1; 54; 70; 40], such as SDEdit [40], balanced authenticity and fidelity by adding noise, while Prompt2Prompt [24] and Plug-and-Play (PNP) [61] enhanced editing through attention mechanisms. Further research, including MasaCtrl [5], Instructpix2pix [4], and InfEdit [69], explored non-rigid editing, specialized image editing models, and rapid editing via consistency sampling. Advances in image inversion and reconstruction [20] have focused on diffusion-based denoising process inversion, categorized into deterministic and non-deterministic sampling [28]. Deterministic methods, such as Null-text inversion using DDIM sampling [41], precisely recover original images but require lengthy optimization; non-deterministic methods, such as DDPM inversion [25] and CycleDiffusion [67], achieve precision by storing variance noise. PnPinv [26] simplifies the process by accurately replacing latent features during denoising, achieving perfect reconstruction but with weaker editability.We propose a framework for creative object synthesis using object textual descriptions for effective fusion and a regularization technique to enhance PnPinv editability.

**Object Composition.** Compositional Text-to-Image synthesis and multi-image subject blending methods [37; 19; 58; 70; 59] aim to create novel images by integrating various concepts, including object interactions, colors, shapes, and attributes. Numerous methodologies [8; 71; 24; 52; 55] have been developed focusing on object combinations, context integration, segmentation, and text descriptions. However, these methods often merely assemble components without effectively melding inter-object relationships, resulting in compositions that, while accurate, lack deeper integration and interaction. This limitation is particularly evident in image editing, where multiple objects in a single image fail to achieve cohesive synthesis. Our method addresses this by harmoniously fusing two objects to create novel entities, thereby enhancing creativity and imagination.

**Semantic Mixing.** The breadth of creativity spans diverse fields, from scientific theories to culinary recipes, driving advancements in AI as highlighted by scholars [3][39] and recent researchers [62] [32]. This creativity has led to significant innovations in AI, particularly through generative models. Creative Adversarial Networks [16] push traditional art boundaries, producing norm-defying works

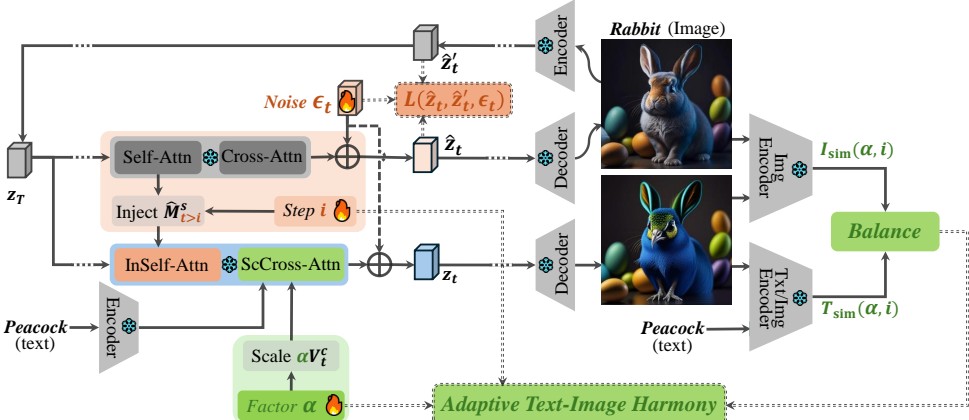

Figure 3: **Framework of our object synthesis** incorporating a scale factor $\alpha$, an injection step $i$ and noise $\epsilon_t$ in the diffusion process. We design a balance loss for optimizing the noise $\epsilon_t$ to balance object editability and fidelity. Using the optimal noise $\epsilon_t$, we introduce an adaptive harmony mechanism to adjust $\alpha$ and $i$, balancing text (*Peacock*) and image (*Rabbit*) similarities.

while maintaining artistic connections. Efforts to adapt AI for novel engineering designs [11] further exemplify this technological creativity. MagicMix [34] introduced semantic mixing task,unlike traditional style transfer methods [73; 60; 10] which blending two concepts into a photo-realistic object while retaining the original image's layout and geometry, but often resulting in biased images and less harmonious fusion. ConceptLab [50] uses diffusion models to generate unique concepts, like new types of pets, but requires time-consuming optimization and struggles to semantically blend real images. Our method operates at the attention layer of diffusion models for harmonious semantic fusion and proposes an adaptive fast search to quickly produce balanced, fused images, ensuring novel and cohesive integration of semantic concepts.

## 3   Methodology

Let $O_I$ and $O_T$ be an object image and an object text, respectively, used as inputs for diffusion models. Our goal is to create a novel object image $O$ by combining $O_I$ with $O_T$ during the diffusion process. To achieve this goal, we develop an adaptive text-image harmony (ATIH) method in our object synthesis framework, as shown in Fig. 3. In subsection 3.1, we introduce a text-image diffusion model with a scale factor $\alpha$, an injection step $i$ and noise $\epsilon_t$. In subsection 3.2, we present to optimize the noise $\epsilon_t$ to balance object editability and fidelity. In subsection 3.3, we propose a simple yet effective ATIH method to adaptively adjust $\alpha$ and $i$ for harmonizing text and image.

### 3.1   Text-Image Diffusion Model (TIDM)

Here, we construct a Text-Image Diffusion Model (TIDM) by utilizing the pre-trained SDXL Turbo [56]. The key components include dual denoising branches: inversion for inverting the input object image, and fusion for fusing the object text and image. Following the latent diffusion model [51], the input latent codes are defined as $z_0 = \mathcal{E}(O_I)$ for object image $O_I$ and $\tau = \mathcal{E}(O_T)$ for object text $O_T$, using a pre-trained image/text encoder $\mathcal{E}(\cdot)$. $\tau_N = \mathcal{E}(O_N)$ denotes as a null-text embedding. The **latent denoising process** is described as follows:

***Inversion Denoising.*** The inversion denoising process predicts the latent code at the previous noise level, $\widehat{z}_{t-1}$, based on the current noisy data $\widehat{z}_t$. This process is defined as:

$$\widehat{z}_{t-1} = \nu_t \widehat{z}_t + \beta_t \epsilon_\theta(\widehat{z}_t, t, \tau) + \gamma_t \epsilon_t, \tag{1}$$

where $\nu_t$, $\beta_t$ and $\gamma_t$ are sampler parameters, $\epsilon_t$ is sampled noise, and $\epsilon_\theta(\widehat{z}_t, t, \tau)$ is a pre-trained U-Net model [56] with self-attention and cross-attention layers. The self-attention is implemented as:

$$\text{Self-Attn}\left(\widehat{Q}_t^s, \widehat{K}_t^s, \widehat{V}_t^s\right) = \widehat{M}_t^s \cdot \widehat{V}_t^s, \quad \widehat{M}_t^s = \text{Softmax}\left(\widehat{Q}_t^s(\widehat{K}_t^s)^T / \sqrt{d}\right), \tag{2}$$

where $\widehat{Q}_t^s$, $\widehat{K}_t^s$ and $\widehat{V}_t^s$ are the query, key and value features derived from the representation $\widehat{z}_t$, and $d$ is the dimension of projected keys and queries. The cross-attention is to control the synthesis process

through the input null-text embedding $\tau_N$, implemented as follows: Cross-Attn $\left(\widehat{Q}_t^c, \widehat{K}_t^c, \widehat{V}_t^c\right) = \widehat{M}_t^c \cdot \widehat{V}_t^c$, where $\widehat{M}_t^c = \text{Softmax}\left(\widehat{Q}_t^c(\widehat{K}_t^c)^T/\sqrt{d}\right)$, $\widehat{Q}_t^c$ is the query feature derived from the output of the self-attention layer, $\widehat{K}_t^c$ and $\widehat{V}_t^c$ are the key and value features derived from $\tau_N$.

***Fusion Denoising.*** Similar to the inversion denoising branch, we redefine the self-attention and cross-attention for easily adjusting the balance between the image latent code $z_t$ and the text embedding $\tau$. The fusion denoising process is redefined as:

$$z_{t-1} = \nu_t z_t + \beta_t \epsilon_\theta(z_t, t, \tau, \alpha, i) + \gamma_t \epsilon_t, \tag{3}$$

where $\nu_t$, $\beta_t$, $\gamma_t$ and $\epsilon_t$ are defined as Eq. (1), and $\epsilon_\theta(z_t, t, \tau, \alpha, i)$ is also the pre-trained U-Net model [56] with injected self-attention and scale cross-attention layers. The **injected self-attention** with an adjustable injection step $i(0 \le i \le T)$ is implemented as:

$$\text{InSelf-Attn}\left(M_t^s, V_t^s\right) = M_t^s \cdot V_t^s, \quad M_t^s = \begin{cases} \widehat{M}_t^s, & \text{if } t > i \\ \text{Softmax}\left(Q_t^s(K_t^s)^T/\sqrt{d}\right), & \text{otherwise} \end{cases}, \tag{4}$$

where $Q_t^s$, $K_t^s$ and $V_t^s$ are the query, key and value features derived from the representation $z_t$. Unlike the approach of injecting $\widehat{K}_t^s$ and $\widehat{V}_t^s$ from Eq. (2) into $K_t^s$ and $V_t^s$ in MasaCtrl [5], we focus on adjusting the injection step $i$ by injecting $\widehat{M}_t^s$ from Eq. (2) into $M_t^s$. The **scale cross-attention** with an adjustable factor $\alpha \in [0, 2]$ is to control the synthesis process through the input text embedding $\tau$, implemented as follows:

$$\text{ScCross-Attn}\left(Q_t^c, K_t^c, V_t^c\right) = M_t^c \cdot \alpha \cdot V_t^c, \quad M_t^c = \text{Softmax}\left(Q_t^c(K_t^c)^T/\sqrt{d}\right), \tag{5}$$

where $Q_t^c$ is the query feature derived from the output of the self-attention layer, $K_t^c$ and $V_t^c$ are the key and value features derived from the text embedding $\tau$. Unlike the non-adjustable scale attention map approach in Prompt-to-Prompt [24], we introduce a factor, $\alpha$, to adjust the value feature. This allows for better balancing of the text and image features, even though they share the same form. Using this fusion denoising process, the generation of a new object image is denoted as $O$.

Following the ReNoise inversion technique [20], based on the denoising Eq. (1) and the approximation $\epsilon_\theta(\widehat{z}_t, t, \tau) \approx \epsilon_\theta(\widehat{z}_{t-1}, t, \tau)$ [14], the **noise addition process** is reformulated as:

$$\widehat{z}_t' = \left(\widehat{z}_{t-1}' - \beta_t \epsilon_\theta(\widehat{z}_t', t, \tau) - \gamma_t \epsilon_t\right)/\nu_t. \tag{6}$$

### 3.2 Balance fidelity and editability by optimizing the noise $\epsilon_t$ in inversion process

In this subsection, our goal is to achieve better fidelity and editability of the object image during the inversion process. We observe that increasing the Gaussian white noise of the denoising latent code $\widehat{z}_{t-1}$ can enhance editability [46], while reducing the difference between the denoising latent code $\widehat{z}_{t-1}$ and the standard path noise code $\widehat{z}_{t-1}'$ in Eq. (6) can improve fidelity [25; 67]. However, these two objectives are contradictory. To address this, we treat the sampling noise $\epsilon_t$ in Eq.(1) as a learnable parameter. We define a reconstructed $\ell_2$ loss between $\widehat{z}_{t-1}$ and $\widehat{z}_{t-1}'$, $\mathcal{L}_r(\epsilon_t) = \|\widehat{z}_{t-1}' - (\nu_t\widehat{z}_t + \beta_t\epsilon_\theta(\widehat{z}_t, t, \tau) + \gamma_t\epsilon_t)\|$, and a KL divergence loss between $\epsilon_t$ and a Gaussian distribution, $\mathcal{L}_n(\epsilon_t) = \text{KL}(q(\epsilon_t)||p(\mathcal{N}(0, I)))$, to simultaneously handle fidelity and editability. Based on Eqs.(1) and (6), we design a balance loss function as follows:

$$\mathcal{L}(\epsilon_t) = |\mathcal{L}_r(\epsilon_t) - \lambda\mathcal{L}_n(\epsilon_t)|, \tag{7}$$

where $\lambda$ represents the weight to balance $\mathcal{L}_r$ and $\mathcal{L}_n$, and in this paper, we set to $\lambda = \frac{\mathcal{L}_r}{\mathcal{L}_n} = 125$. Since the parameter $\epsilon_t$ is sampled from a standard Gaussian distribution during the noise addition process, $\mathcal{L}_n$ is used solely to balance $\mathcal{L}_r$ and its gradient is not computed for optimization.

### 3.3 Text-image harmony by adaptively adjusting injection step $i$ and scale factor $\alpha$

Using the optimal noise $\epsilon_t$, a fused object image $O(\alpha, i)$ can be generated by the TIDM with an initial scale factor $\alpha_0 = 1$ and injection step $i_0 = \lfloor T/2 \rfloor$ from the input object image $O_I$ and object text

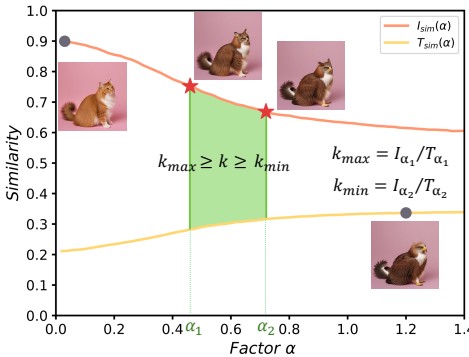
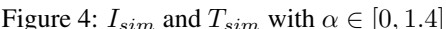

Figure 4: $I_{sim}$ and $T_{sim}$ with $\alpha \in [0, 1.4]$.

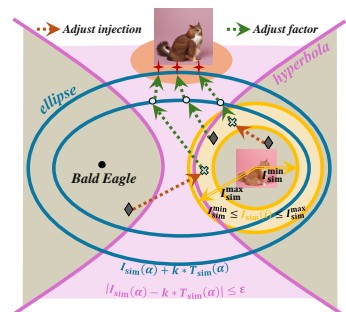

Figure 5: The adjusted process of our ATIH with three initial points and $\varepsilon = I_{\text{sim}}(\alpha) + k \cdot T_{\text{sim}}(\alpha) - F(\alpha)$.

$O_T$. Here, we adaptively adjust $\alpha \in [0, 2]$ and $i$ ($0 \leq i \leq T$) by introducing an Adaptive Text-Image Harmony (ATIH) method. We denote the similarity between the image $O_I$ and the fused image $O(\alpha, i)$ as $I_{\text{sim}}(\alpha, i) = d(O_I, O(\alpha, i))$, and the similarity between the text $O_T$ and the fused image $O(\alpha, i)$ as $T_{\text{sim}}(\alpha, i) = d(O_T, O(\alpha, i))$, where $d(\cdot, \cdot)$ represents the similarity distance between text/image and image. In this paper, we compute the similarities $I_{\text{sim}}(\alpha, i)$ and $T_{\text{sim}}(\alpha, i)$ using the DINO features [44] and the CLIP features [48], respectively, based on a cosine distance $d$. Our key idea is to balance and maximize both $I_{\text{sim}}(\alpha, i)$ and $T_{\text{sim}}(\alpha, i)$ for optimal text-image fusion.

**Adjust injection step $i$ to balance fidelity and editability.** Before achieving the idea, we first enable the object image to be smoothly editable by adjusting the injection step $i$ in the injected self-attention. We denote $I_{\text{sim}}(i) = I_{\text{sim}}(\alpha_0, i)$ for for convenience. In the inversion process, it is generally observed that more injections lead to less editability. When all injections are applied ($i = T$), an ideal fidelity is achieved. We observe that when $I_{\text{sim}}(i) < I_{\text{sim}}^{\min}$, the fused image deviates significantly from the input image, resulting in a loss of fidelity. Conversely, when $I_{\text{sim}}(i) > I_{\text{sim}}^{\max}$, the fused image is too similar to the input image, resulting in no editability. To balance fidelity and editability, $I_{\text{sim}}(i)$ must satisfy $I_{\text{sim}}^{\min} \leq I_{\text{sim}}(i) \leq I_{\text{sim}}^{\max}$, in Fig. 5. Therefore, initializing $i = \lfloor T/2 \rfloor$, $i$ is adjusted as follows:

$$i = \begin{cases} i - 1, & I_{\text{sim}}(i) < I_{\text{sim}}^{\min} \\ i, & I_{\text{sim}}^{\min} \leq I_{\text{sim}}(i) \leq I_{\text{sim}}^{\max} \\ i + 1, & I_{\text{sim}}(i) > I_{\text{sim}}^{\max} \end{cases}, \tag{8}$$

where $I_{\text{sim}}^{\min}$ and $\text{sim}^{\max}$ are set to $0.45$ and $0.85$ in this paper, respectively, based on observations from Fig. 17. After using Eq. (8), this adaptive approach can obtain an injection step $i^*$ to smooth the fusion process while maintaining a balance between fidelity and editability. Fixing the injection step $i = i^*$, next we use abbreviations, $I_{\text{sim}}(\alpha) = I_{\text{sim}}(\alpha, i^*)$ and $T_{\text{sim}}(\alpha) = T_{\text{sim}}(\alpha, i^*)$.

**Adaptively adjust the scale factor $\alpha$ for harmonizing text and image.** To implement our key idea, we design an exquisite score function with $\alpha$ as:

$$\max_{\alpha} F(\alpha) := \underbrace{I_{\text{sim}}(\alpha) + k \cdot T_{\text{sim}}(\alpha)}_{\text{maximize similarities (ellipse)}} - \beta \underbrace{|I_{\text{sim}}(\alpha) - k \cdot T_{\text{sim}}(\alpha)|}_{\text{balance similarities (hyperbola)}}, \tag{9}$$

where $\beta$ is a weighting factor, and the parameter $k$ is introduced to mitigate inconsistencies in scale between high $I_{\text{sim}}(\alpha)$ and low $T_{\text{sim}}(\alpha)$ due to differences in text and image modalities, ensuring their scale balance. As shown in Fig. 4, $I_{\text{sim}}(\alpha)$ decreases and $T_{\text{sim}}(\alpha)$ increases as $\alpha$ increases, and vice versa. Based on these observations, we set $k = 2.3$ and $\beta = 1$ in this paper.

In Eq. (9), the left-hand side represents the sum of the text and image similarities, forming an ellipse, while the right-hand side represents the absolute value of the difference between the text and image similarities, forming a hyperbola. A larger sum value indicates that the generated image integrates more information from the input text and image. Conversely, a smaller absolute value signifies a better balance between the text and image similarities. Additionally, given that $I_{\text{sim}}(\alpha) \in [0, 1]$ and $T_{\text{sim}}(\alpha) \in [0, 1]$, their sum is greater than or equal to the absolute value of their difference, leading to $F(\alpha) \geq 0$. Therefore, our objective is to maximize $F(\alpha)$ to simultaneously enhance and balance both $I_{\text{sim}}(\alpha)$ and and $T_{\text{sim}}(\alpha)$. Maximizing $F(\alpha)$ is easily implemented by the Golden Section Search [47] algorithm, and we get the optimal $\alpha^*$. Fig. 5 depicts a schematic diagram to adjust both ii and $\alpha$. Overall, our **novel object synthesis**, detailed in **Algorithm 1**, is presented in Appendix F.

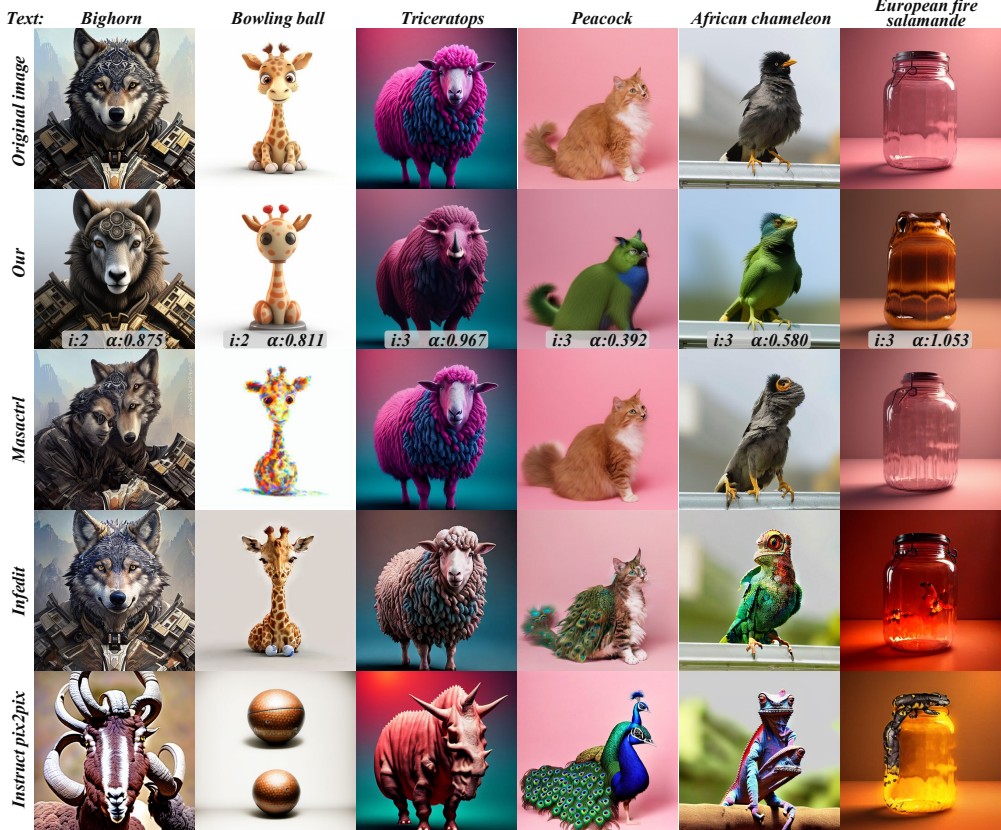

Figure 6: **Comparisons with different image editing methods.** We observe that InfEdit [69] MasaCtrl [5] and InstructPix2Pix [4] struggle to fuse object images and texts, while our method successfully implements new object synthesis, such as *bowling ball-fawn* in the second row.

## 4 Experiments

### 4.1 Experimental Settings

**Datasets.** We constructed an object text-image fusion (OTIF) dataset consisting of 1,800 text-image pairs, derived from 60 texts and 30 images in Appendix C. Images, selected from various classes in PIE-bench [26], include 20 animal and 10 non-animal categories. Texts were chosen from the 1,000 classes in ImageNet [53], with ChatGPT [42] filtering out 40 distinct animals and 20 non-animals.

**Details.** We implemented our method on SDXLturbo [56] only taking ***ten seconds***. For image editing, we set the source prompt $p_s$ as an empty string "Null" and the target prompt $P_t$ as the target object class name. During sampling, we used the Ancestral-Euler sampler [28] with four denoising steps. All input images were uniformly scaled to $512 \times 512$ pixels to ensure consistent resolution in all the experiments. Our experiments were conducted using two NVIDIA GeForce RTX 4090 GPUs.

**Metrics.** To comprehensively evaluate the performance of our method, we employed four key metrics: aesthetic score (AES) [57], CLIP text-image similarity (CLIP-T) [48], Dinov2 image similarity (Dino-I) [44], and human preference score (HPS) [68]. Following the Eq. (9), $F$score and balance similarities ($B$sim) with $k = 2.3$ are used to measure the text-image fusion effect.

### 4.2 Main Results

We conducted a comprehensive comparison of our ATIH model with three image-editing models (*i.e.*, MasaCtrl [5], InfEdit [69], and InstructPix2pix [4]), two mixing models (*i.e.*, MagicMix [34] and ConceptLab [50]), and ControlNet [72]. Notably, MagicMix and ConceptLab share a similar objective with ours to fuse object text/image, while ConceptLab only accepts two text prompts as its inputs. Due to no available code for MagicMix, we utilized its unofficial implementation [13].

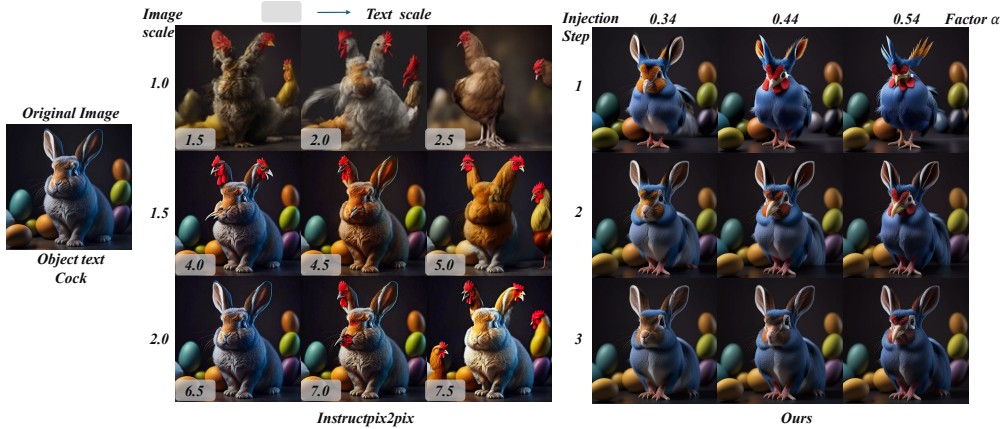

Figure 7: Comparisons with InstructPix2Pix [4] using image/text strength variations.

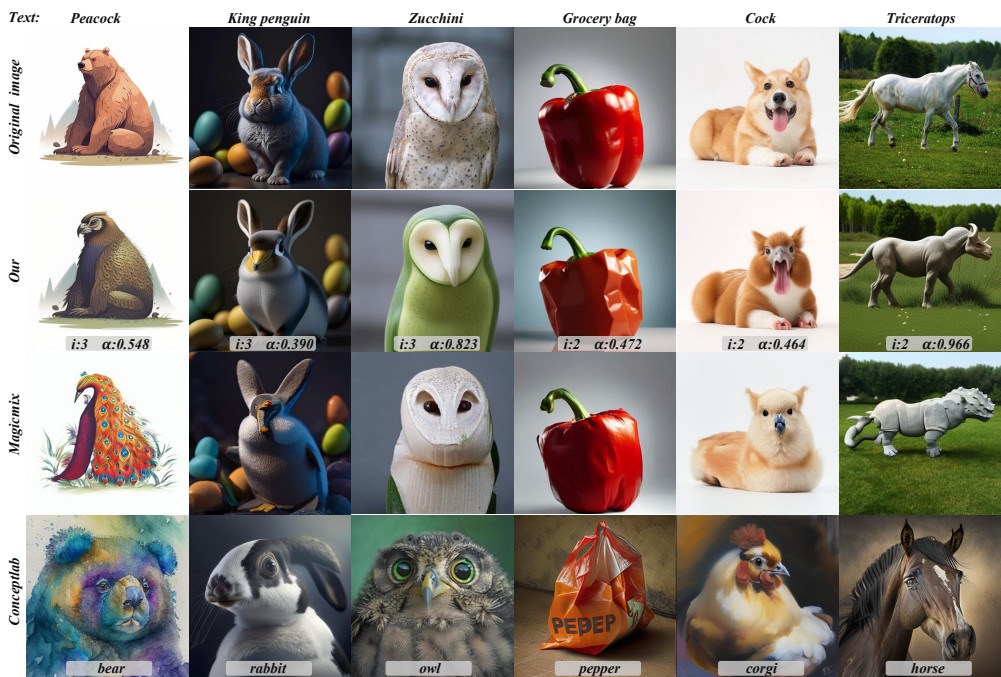

Figure 8: **Comparisons with different creative mixing methods.** We observe that our results surpass those of MagicMix [34]. For ConceptLab [50], we exclusively examine its fusion results without making good or bad comparisons, as it is a distinct approach to creative generation.

**Comparisons with image-editing methods.** For a fair comparison, we uniformly set the editing text prompt in all methods as *a photo of an {image category} creatively fused with a {text category}* to achieve the fusion of two objects. Fig. 6 visualizes some combinational objects, with additional results available in Appendix H. Our observations are as follows: Firstly, MasaCtrl and InfEdit generally preserve the original image's details better during editing, as seen in examples like *sheep-triceratops*. In contrast, InstructPix2Pix tends to alter the image more significantly, making it closer to the edited text description. Secondly, different methods exhibit varying degrees of distortion when fusing two objects during the image editing process. For instance, in the case of *African chameleon-bird*, our method performs better by minimizing distortions and maintaining the harmony and high quality of the image. Thirdly, our method shows significant advantages in enhancing the editability of images. For the *European fire salamander-glass jar* example, other methods often result in only color changes and slight deformations, failing to effectively merge the two objects. In contrast, our method harmoniously integrates the colors and shapes of both the glass jar and the European fire salamander, significantly improving the editing effect and operability. Specially, Fig. 7 shows the results of InstructPix2Pix with manually adjusted image strengths (1.0, 1.5, 2.0) and text strengths (ranging

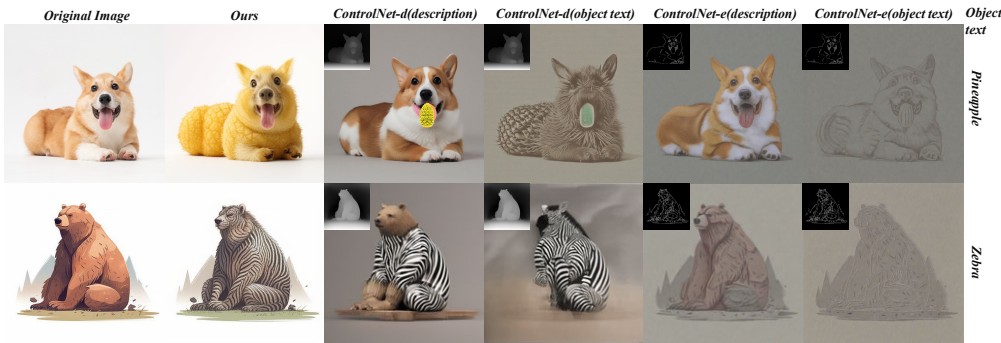

Figure 9: Comparisons with ControlNet-depth and ControlNet-edge [72] using a description that "A photo of an {*object image*} creatively fused with an {*object text* }".

from 1.5 to 7.5). At optimal settings of image strength 1.5 and text strength 5.0, InstructPix2Pix produced its best fusion, though some results were unnatural, like replacing the rabbit's ears with a rooster's head. In contrast, our method created novel and natural combinations of the rabbit and rooster by automatically achieving superior visual synthesis without manual adjustments.

**Comparisons with the mixing methods.** Fig. 8 illustrates the results of text-image object synthesis. We observe that both MagicMix and ConceptLab tend to overly bias towards one class, such as *zucchini-owl* and *corgi-cock*. Their generated images often lean more towards one category. In contrast, our method achieves a more harmonious balance between the features of the two categories. Moreover, the fusion images produced by MagicMix frequently exhibit insufficiently smooth feature blending. For instance, in the fusion of a rabbit and an emperor penguin, the rabbit's facial features nearly disappear. Conversely, our method seamlessly merges the facial features of both the penguin and the rabbit in the head region, preserving the main characteristics of each.

**Comparisons with ControlNet.** We rigorously compared our method with ControlNet to assess their performance in complex text-image fusion tasks, as shown in Fig. 9. Our results highlight notable differences: ControlNet preserves structure well from depth or edge maps but struggles with semantic integration, especially with complex prompts, often failing to achieve seamless blending. In contrast, our method leverages full RGB features, including color and texture, alongside structural data.

Table 1: Quantitative comparisons on our TIF dataset.

| Models | DINO-I↑ [44] | CLIP-T↑ [48] | AES ↑ [57] | HPS↑ [68] | $F$score↑ | $B$sim↓ |
|---|---|---|---|---|---|---|
| **Our ATIH** | 0.756 | 0.296 | **6.124** | **0.383** | **1.362** | **0.075** |
| MagicMix [34] | 0.587 | 0.328 | 5.786 | 0.373 | 1.174 | 0.167 |
| InfEdit [69] | **0.817** | 0.255 | 6.080 | 0.367 | 1.173 | 0.230 |
| MasaCtrl [5] | 0.815 | 0.234 | 5.684 | 0.343 | 1.077 | 0.277 |
| InstructPix2Pix [4] | 0.384 | **0.394** | 5.881 | 0.375 | 0.768 | 0.522 |

Table 2: $H$-statistics (↑) ($P$-value (↓)) between our ATIH and other methods under different metrics.

| Methods | DINO-I [44] | CLIP-T [48] | AES [57] | HPS [68] | $F$score | $B$sim |
|---|---|---|---|---|---|---|
| MagicMix [34] | 665.20 ($1.10e^{-146}$) | 248.15 ($6.58e^{-56}$) | 433.00 ($3.61e^{-96}$) | 232.1 ($1.45e^{-08}$) | 633.89 ($7.13e^{-140}$) | 792.72 ($2.06e^{-174}$) |
| InfEdit [69] | 402.36 ($1.68e^{-89}$) | 477.31 ($8.22e^{-106}$) | 3.70 ($5.45e^{-02}$) | 114.02 ($1.29e^{-26}$) | 504.53 ($9.81e^{-112}$) | 917.99 ($1.20e^{-201}$) |
| MasaCtrl [5] | 404.87($4.81e^{-90}$) | 943.37($3.67e^{-207}$) | 277.80 ($2.27e^{-62}$) | 654.62 ($2.21e^{-144}$) | 991.48 ($1.28e^{-217}$) | 1183.59 ($2.25e^{-259}$) |
| InstructPix2Pix [4] | 1565.18 (0.000000) | 1891.69 (0.000000) | 268.57 ($2.32e^{-60}$) | 39.63 ($3.06e^{-10}$) | 1421.64 ($4.18e^{-311}$) | 1997.67(0.000000) |

**Quantitative Results.** Table 1 displays the quantitative results, illustrating that our method achieves state-of-the-art performance in AES, HPS, $F$score and $B$sim, surpassing other methods. These results indicate that our approach excels in enhancing the visual appeal and artistic quality of images, while also aligning more closely with human preferences and understanding in terms of object fusion. Moreover, when dealing with text-image inconsistencies at scale $k$=2.3, our method achieves superior text-image similarity and balance, demonstrating superior fusion capability. Despite achieving the best DINO-I and CLIP-T scores under inconsistencies, InfEdit and InstructPix2Pix perform worse than our method in terms of AES, HPS, $F$score and $B$sim, and their visual results remain sub-optimal. These inconsistencies ultimately lead to the failure of integrating object text and image. In contrast, our approach achieves a better text-image balance similarities. Furthermore, Table 2 presents the $H$-statistics [30] and $P$-values [66] assessing the statistical significance of performance differences between our ATIH and other methods across various metrics. Compared to Instructpix2pix, for

instance, our method shows significant differences, with $H$-statistics of 268.57 for AES and 39.63 for HPS, indicating potential improvements in both aesthetic quality and human preference scoring.

**User Study.** We conducted two user studies to assess intuitive human perception of results presented in Table 3, Table 4, and Appendix G. Each participant evaluated 6 image-editing sets and 6 fusion sets. In total, these studies garnered 570 votes from 95 participants. Our method received the highest ratings in both studies, capturing 74.03% and 79.47% of the total votes, respectively. Among the image-editing methods, InfEdit [69] garnered 14.7% of votes for its superior editing performance, while InstructPix2Pix [4] and MasaCtrl [5] received only 8% and 2.8%, respectively. In the fusion category, ConceptLab [50] received 12.28% of votes, while MagicMix [34] received 8%.

Table 3: User study with image editing methods.

| Models | Our ATIH | MasaCtrl[5] | InstructPix2Pix[4] | InfEdit[69] |
|---|---|---|---|---|
| Vote ↑ | **422** | 16 | 48 | 84 |

Table 4: User study with mixing methods.

| Models | Our ATIH | MagicMix[34] | ConceptLab[50] |
|---|---|---|---|
| Vote ↑ | **453** | 47 | 70 |

### 4.3 Parameter Analysis and Ablation Study

**Parameter analysis.** Our primary parameters include $\lambda$ in (7), $I_{\text{sim}}^{\min}$ and $I_{\text{sim}}^{\max}$ in (8), and $k$ in (9). $\lambda = \frac{\mathcal{L}_r}{\mathcal{L}_n}$ balances the editability and fidelity of our model. We determined the specific value through personal observation combined with the changes in AES, CLIP-T, and Dino-I values at different $\lambda$ settings. Ultimately, we set $\lambda$ to 125. To address the inconsistency between image similarity and text similarity scales, we approximated the scale $k$. Initially, we measured the variations in image similarity and text similarity with changes in $\alpha$, and identified the balanced similarity regions in the fusion results. As shown in Fig. 4, the optimal range for $k$ was found to be between [0.21, 0.27]. Based on these observations and experimental experience, we ultimately set $k$ to 0.23. As shown in Fig. 17 of Appendix D, we observe that when the similarity between the image and the original exceeds 0.85, the images become too similar, making edits with different class texts less effective and necessitating a decrease in $i$. Conversely, when the similarity is below 0.45, the images overly favor the text, making them excessively editable, requiring an increase in injection steps. Therefore, we set $I_{\text{sim}}^{\min}$ to 0.45 and $I_{\text{sim}}^{\max}$ to 0.85. More discussions are provided in Appendix D.

**Ablation Study.** In Figs. 10 and 18 in Appendix E, we visualize the results with and without the balance loss in Eq. (7), the adaptive injection ii in Eq. (8), and the adaptive selection $\alpha$ in Eq. (9) within our object synthesis framework. Pn-Pinv, used for direct inversion and prompt editing, resulted in some distortion and blurriness. Compared to PnPinv, the balance loss

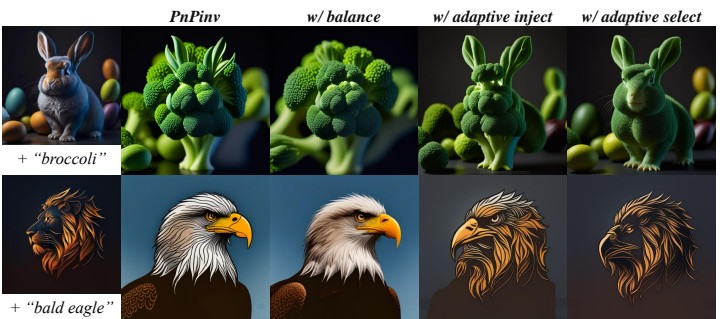

Figure 10: Ablation study of the balance loss, adaptive injection ii and adaptive selection $\alpha$ from the third column to the fifth column.

significantly enhances image fidelity, improving details, textures, and editability. The adaptive injection enables a smooth transition from Corgi to Fire Engine in Fig. 18. Without this injection, the transformation is too abrupt, lacking a seamless fusion process. Finally, the adaptive selection achieves a balanced image that harmoniously integrates the original and target features. ***Note that for limitations, please refer to Appendix B.***

## 5   Conclusion

In this paper, we explored a novel object synthesis framework that fuses object texts with object images to create unique and surprising objects. We introduced a simple yet effective difference loss to optimize sampling noise, balancing image fidelity and editability. Additionally, we proposed an adaptive text-image harmony module to seamlessly integrate text and image elements. Extensive experiments demonstrate that our framework excels at generating a wide array of impressive object combinations. This capability is particularly advantageous for crafting innovative and captivating animated characters in the entertainment and film industry. Broader impact, please see Appendix A.

## Acknowledgements

This work was partially supported by the National Science Fund of China, Grant Nos. 62072242 and 62361166670. We sincerely appreciate the valuable feedback provided by the anonymous reviewers.

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

## A  Broader Impact

Our model's capability to fuse images and text to generate new and creative object images holds significant potential across various fields, including entertainment, design, and education. However, it also raises important considerations regarding content safety and ethical use. In particular, if the input image or text contains inappropriate or offensive material, the generated images may similarly be inappropriate, leading to potentially unpleasant experiences for users.

To mitigate these risks, it is crucial to implement robust NSFW (Not Safe For Work) content detection mechanisms. While existing methods can address some cases of inappropriate content, we acknowledge the need for continuous improvement in this area. As part of our future work, we will incorporate advanced NSFW checking models to ensure the generated content adheres to safety standards and ethical guidelines. This proactive approach aims to safeguard users and promote responsible use of our image generation technology.

## B  Limitation

Our method relies on the semantic correlation between the original and transformed content within the diffusion feature space. When the semantic match between two categories is weak, our method tends to produce mere texture changes rather than deeper semantic transformations. This limitation suggests that our approach may struggle with transformations between categories with weak semantic associations. Future work could focus on enhancing semantic matching between different categories to improve the generalizability and applicability of our method.

There are still some failure cases in our model, as shown in Fig. 11. These failures can be categorized into two types. The first row illustrates that when the content of the image is significantly different from the text prompt, the changes become implicit. The second row demonstrates that in certain cases, our adaptive function results in changes that only affect the texture of the original image. In our future work, we will investigate these situations further and analyze the specific items that do not yield satisfactory results.

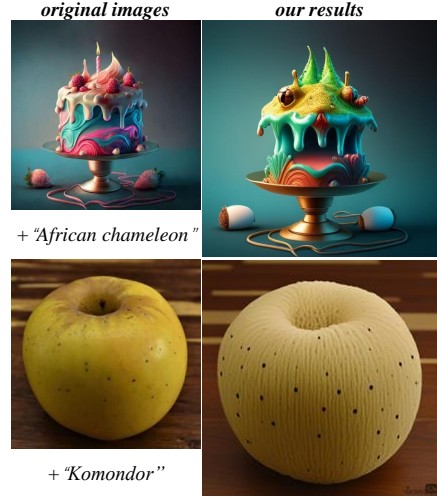

Figure 11: Failure results of our ATIH model.

## C  Text and Image Categories.

We selected 60 texts, as detailed in Table 5, and categorized them into 7 distinct groups. The 30 selected images are shown in Fig.12, with each image corresponding to similarly categorized texts, as outlined in Table 6. Our model is capable of fusing content between any two categories, showcasing its strong generalization ability.

Table 5: List of Text Items by Object Category.

| Category | Items |
| --- | --- |
| Mammals | kit fox, Siberian husky, Australian terrier, badger, Egyptian cat, cougar, gazelle, porcupine, sea lion, bison, komondor, otter, siamang, skunk, giant panda, zebra, hog, hippopotamus, bighorn, colobus, tiger cat, impala, coyote, mongoose |
| Birds | king penguin, indigo bunting, bald eagle, cock, ostrich, peacock |
| Reptiles and Amphibians | Komodo dragon, African chameleon, African crocodile, European fire salamander, tree frog, mud turtle |
| Fish and Marine Life | anemone fish, white shark, brain coral |
| Plants | broccoli, acorn |
| Fruits | strawberry, orange, pineapple, zucchini, butternut squash |
| Objects | triceratops, beach wagon, beer glass, bowling ball, brass, airship, digital clock, espresso maker, fire engine, gas pump, grocery bag, harp, parking meter, pill bottle |

Table 6: Original Object Image Categories.

| Category | Items |
|---|---|
| Mammals | Sea lion, Dog (Corgi), Horse, Squirrel, Sheep, Mouse, Panda, Koala, Rabbit, Fox, Giraffe, Cat, Wolf, Bear |
| Birds | Owl, Duck, Bird |
| Insects | Ladybug |
| Plants | Tree, Flower vase |
| Fruits and Vegetables | Red pepper, Apple |
| Objects | Cup of coffee, Jar, Church, Birthday cake |
| Human | Man in a suit |
| Artwork | Lion illustration, Deer illustration, Twitter logo |

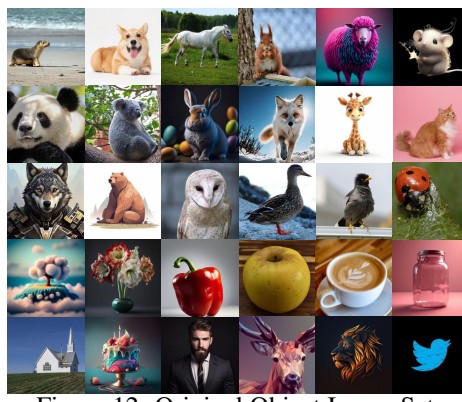

Figure 12: Original Object Image Set.

# D Parameter Analysis.

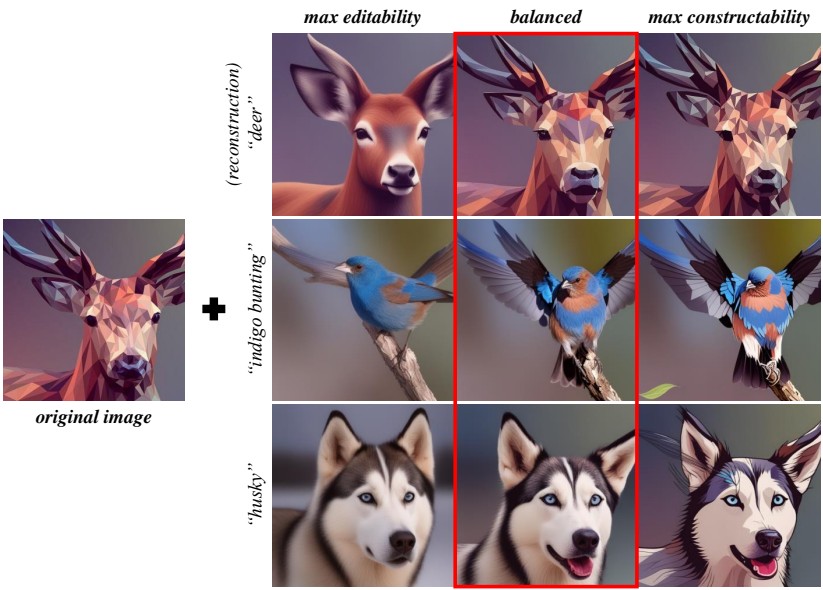

Figure 13: Image variations under different $\lambda$ values. The first row displays the reconstructed images. The middle and bottom rows show the results of editing with different prompts, demonstrating variations in maximum editability, a balanced approach, and maximum constructability

**Analysis of $\lambda$.** Here, we provide a detailed explanation of the determination of $\lambda$. As shown in Fig. 13, we use the ratio $\lambda = \frac{L_r}{L_n}$ to balance editability and fidelity. We iteratively adjust this ratio in the range of $[0, 400]$ with intervals of 10, measuring the Dino-I score between the reconstructed and original images, as well as the CLIP-T and AES scores for images directly edited with the inverse latent values at different ratios. These experiments were conducted on the class fusion dataset, using fusion

Table 7: Quantitative comparison results with different $\lambda$.

| $\lambda$ | AES ↑ | CLIP-T ↑ | Dino-I ↑ |
|---|---|---|---|
| 0 | 6.116 | 0.413 | 0.927 |
| 125 | **6.153** | 0.417 | 0.902 |
| 260 | 6.012 | 0.419 | 0.760 |

text for direct image editing. Figs. 14, 15, and 16 indicate that as the ratio increases, image editability improves, peaking at a ratio of around 260, but with a decrease in quality. At a ratio of 125, both image fidelity and the AES score achieve an optimal balance. Therefore, we set $\lambda$ to 125.

**Analysis of $k$.** The experimental analysis of parameter $k$ was conducted using sdxlturbo as the base model. The range for $i$ was set to $[0, 4]$, and for each value of $i$, $\alpha$ was iterated from 0 to 2.2 in steps of 0.02 to observe changes in the fused image. The averaged experimental results produced a smooth curve, as shown in Fig.4. Based on these observations, the optimal range for $k$ was determined to be between $[2.1, 2.7]$. In our experiments, we set the value of $k$ to 2.3.

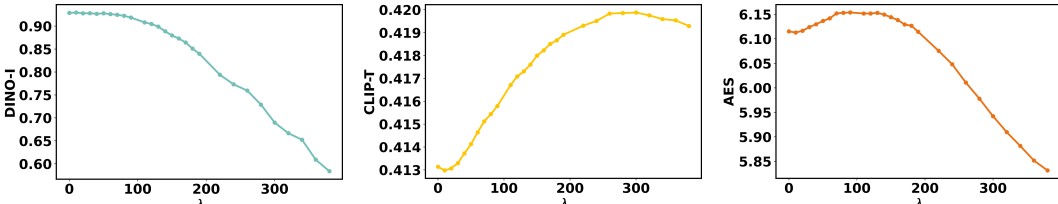

Figure 14: Dino-I changing with $\lambda$

Figure 15: CLIP-T score changing with $\lambda$

Figure 16: AES changing with $\lambda$

**Analysis of $I_{\text{sim}}^{\min}$ and $I_{\text{sim}}^{\max}$.** As shown in Fig. 17, we visualized several specific node images generated during the variation of different $\alpha$ factor values. When the image similarity with the original image exceeds 0.85, the images become overly similar. For example, in the dog-zebra fusion experiment, the dog's texture remains largely unchanged, and no zebra features are visible. Conversely, when the image similarity falls below 0.45, the images overly conform to the text description. In this case, the entire head of the image turns into a zebra, representing an over-transformation phenomenon. Based on these observations, we set the minimum similarity threshold $I_{\text{sim}}^{\min}$ to 0.45 and the maximum similarity threshold $I_{\text{sim}}^{\max}$ to 0.85. This range helps us achieve a good balance between retaining original image information and integrating text features.

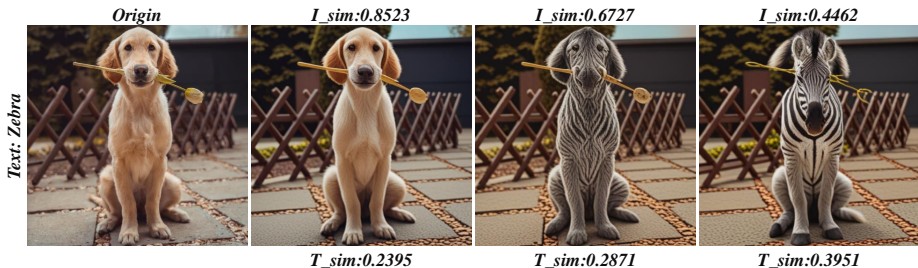

Figure 17: Illustrates the visual results of images at different similarity levels.

# E   Ablation Study.

We present another set of ablation study results in Fig. 18, where the two rows represent the cases without (w/o) and with (w) attention projection. The input image is a Corgi, and the text is Fire engine. The output images display the different transformations as $\alpha$ varies. The top row shows the abrupt change in appearance without attention projection, resulting in a sudden transition from a Corgi to a fire engine. In contrast, with attention projection (bottom row), the change is smoother, achieving the desired blending result in the middle.

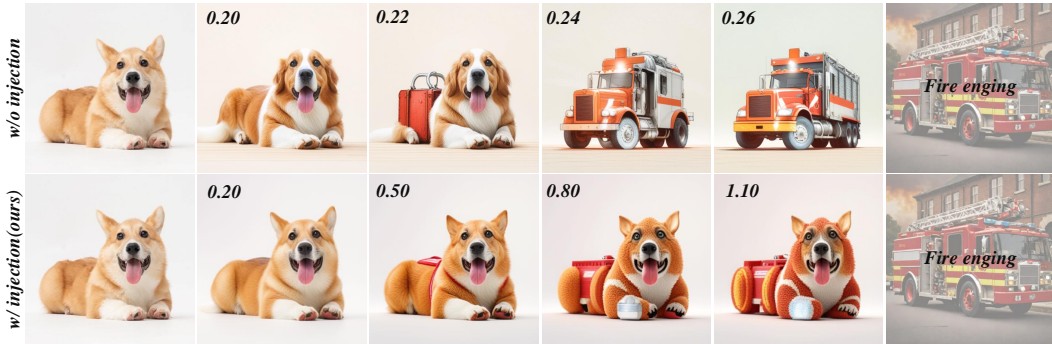

Figure 18: Results changing in Iteration w/ and w/o attention injection.

# F   Algorithm.

Overall, our **novel object synthesis** comprises three key components: optimizing the noise $\epsilon_t$ through a balance of fidelity and editability loss, adaptively adjusting the injection step $i$, and dynamically modifying the factor

---

**Algorithm 1** Novel Object Synthesis

---

1: **Input:** An initial image latent $z_0$, a target prompt $O_T$, the number of inversion steps $T$, inject step $i$, sampled noise $\epsilon_t$, scale factor $\alpha$, $F(\alpha)$ is Eq.(9)
2: **Output:** Object Synthesis $O$
3: $\{z_T, \cdots, \widehat{z}'_{t-1}, \cdots, z_0\} \leftarrow \text{scheduler\_inverse}(z_0)$
4: **for** $t = 1$ **to** $T$ **do**
5:     $\widehat{z}_{t-1} \leftarrow \text{step}(\widehat{z}_t)$
6:     $\epsilon_{\text{all}}[t] \leftarrow \text{Balance-fidelity-editability}(\widehat{z}_{t-1}, \widehat{z}'_{t-1}, \widehat{z}_t, \epsilon_t)$
7: **end for**

8: $i_{\text{init}} \leftarrow T/2$
9: $i_{\text{final}} \leftarrow \text{Adjust-Inject}(z_T, \epsilon_{\text{all}}, O_T, i_{\text{init}})$
10: $\alpha_{\text{good}} \leftarrow \text{Golden-Section-Search}(F, \alpha_{\text{min}}, \alpha_{\text{max}})$
11: $O \leftarrow \text{DM}(z_T, \epsilon_{\text{all}}, O_T, i_{\text{final}}, \alpha_{\text{good}})$
12: **return** $O$

---

13: **function** BALANCE-FIDELITY-EDITABILITY$(\widehat{z}_{t-1}, \widehat{z}_{t-1}, \widehat{z}'_{t-1}, \epsilon_t)$
14:     **while** $\mathcal{L}_{\text{r}}/\mathcal{L}_{\text{n}} > \lambda$ **do**
15:        $\epsilon_t \leftarrow \epsilon_t - \nabla_{\epsilon_t} \mathcal{L}_{\text{r}}(\widehat{z}_{t-1}, \widehat{z}'_{t-1}, \epsilon_t, \widehat{z}_t)$
16:     **end while**
17:     **return** $\epsilon_t$
18: **end function**

---

19: **function** GOLDEN-SECTION-SEARCH$(F, a, b)$
20:     $\phi \leftarrow \frac{1+\sqrt{5}}{2}$                                         ▷ Golden ratio
21:     $c \leftarrow b - \frac{b-a}{\phi}$
22:     $d \leftarrow a + \frac{b-a}{\phi}$
23:     **while** $|b - a| > \epsilon$ **do**
24:        **if** $f(c) < f(d)$ **then**
25:           $b \leftarrow d$
26:        **else**
27:           $a \leftarrow c$
28:        **end if**
29:        $c \leftarrow b - \frac{b-a}{\phi}$
30:        $d \leftarrow a + \frac{b-a}{\phi}$
31:     **end while**
32:     **return** $\frac{b+a}{2}$
33: **end function**

---

34: **function** ADJUST-INJECT$(z_T, \epsilon_{all}, i, O_T)$
35:     $ite \leftarrow 0$
36:     **while** $iter < \frac{T}{2}$ **do**
37:        $I_{\text{sim}} \leftarrow \text{model}_{I_{\text{sim}}}(z_T, \epsilon_{all}, i, O_T)$
38:        **if** $I_{\text{sim}} < I_{\text{sim}}^{\text{min}}$ **then**
39:           $i \leftarrow i + 1$
40:        **else if** $I_{\text{sim}}^{\text{min}} \leq I_{\text{sim}} \leq I_{\text{sim}}^{\text{max}}$ **then**
41:           $i \leftarrow i$
42:           **break**
43:        **else**
44:           $i \leftarrow i - 1$
45:        **end if**
46:        $iter \leftarrow iter + 1$
47:     **end while**
48:     **return** $i$
49: **end function**

---

$\alpha$. These processes are detailed in Algorithm 1. Additionally, we utilize the Golden Section Search method to identify an optimal or sufficiently good value for $\alpha$ that maximizes the score function $F(\alpha)$ in Eq. (9). This approach operates independent of the function's derivative, enabling rapid iteration towards achieving optimal harmony. The key steps of the Golden Section Search algorithm are outlined as follows:

$$\alpha_1 = b - \frac{b - a}{\phi}, \quad \alpha_2 = a + \frac{b - a}{\phi},$$

where $\phi$ (approximately 1.618) is the golden ratio, and $a$ and $b$ are the current search bounds for $\alpha$. During each iteration, we compare $F(\alpha_1)$ and $F(\alpha_2)$, and adjust the search range accordingly:

$$\text{if } F(\alpha_1) > F(\alpha_2) \text{ then } b = \alpha_2 \text{ else } a = \alpha_1.$$

This process continues until the length of the search interval $|b - a|$ is less than a predefined tolerance, indicating convergence to a local maximum.

# G   User Study.

In this section, we delve into our two user studies in greater detail. The image results are illustrated in Figs. 6 and 8, while the outcomes of the user studies for both tasks are presented in Figs. 19 and 20. In total, we collected 570 votes from 95 participants across both studies. The specific responses for each question are detailed in Tables 8 and 9.

Notably, for the fourth question in the user study corresponding to our editing method, the example of peacock and cat fusion is shown in Fig.6, the number of votes for InfEdit [69] slightly exceeded ours. However, upon examining the image results, it becomes evident that their approach leans towards a disjointed fusion, where one half of an object is spliced with the corresponding half of another object, rather than directly generating a new object as our method does.

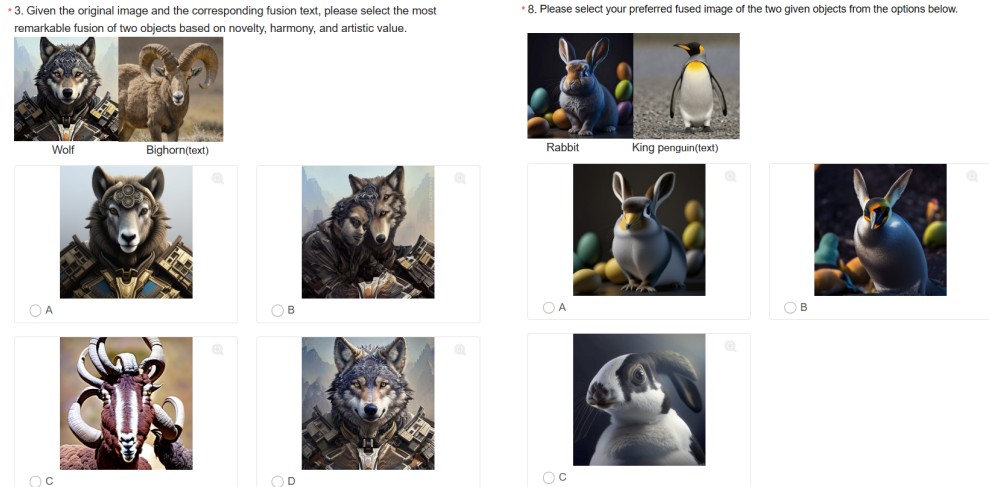

Figure 19: An example of a user study comparing various image-editing methods.

Figure 20: An example of a user study comparing various mixing methods.

Table 8: User study with image editing methods.

| options(Models) / image-prompt | A(Our ATIH) | B(MasaCtrl) | C(InstructPix2Pix) | D(InfEdit) |
|---|---|---|---|---|
| glass jar-salamander | 77.89 % | 1.05% | 16.84% | 4.21% |
| giraffe-bowling ball | 89.74 % | 2.11% | 2.11% | 6.32% |
| wolf-bighorn | 84.21 % | 1.05% | 10.53% | 4.21% |
| cat-peacock | 40 % | 3.16% | 5.26% | 51.58% |
| sheep-triceraptors | 78.95 % | 3.16% | 11.58% | 6.32% |
| bird-African chameleon | 73.68 % | 6.32% | 4.21% | 15.79% |

Table 9: User study with mixing methods.

| options(Models)
(prompt)
image-prompt | A(Our ATIH) | B(MagicMix) | C(ConceptLab) |
|---|---|---|---|
| Dog-white shark | 81.05% | 2.11% | 16.84% |
| Rabbit-king penguin | 83.16% | 11.58% | 5.26% |
| horse-microwave oven | 71.58% | 9.47% | 18.95% |
| camel-candelabra | 86.32% | 6.32% | 7.37% |
| airship-espresso maker | 71.58% | 11.58% | 16.84% |
| jeep-anemone fish | 83.16% | 8.42% | 8.42% |

## H   More results.

In this section, we present additional results from our model. Fig. 21 showcases further generation results using our ATIH model. We experimented with four different images, each edited with four distinct text prompts. Fig. 22 provides further examples showcasing the effectiveness of our method in complex text-driven fusion tasks. Specifically, our approach excels in extreme cases by accurately extracting prominent features, such as color and basic object forms, from detailed textual descriptions. For instance, Fig. 22 shows a well-defined edge structure for the fawn image and the text 'Green triceratops with rough, scaly skin and massive frilled head.' Additionally, Fig. 23 illustrates our model's versatility with multiple prompts, emphasizing its capability for continuous editing.

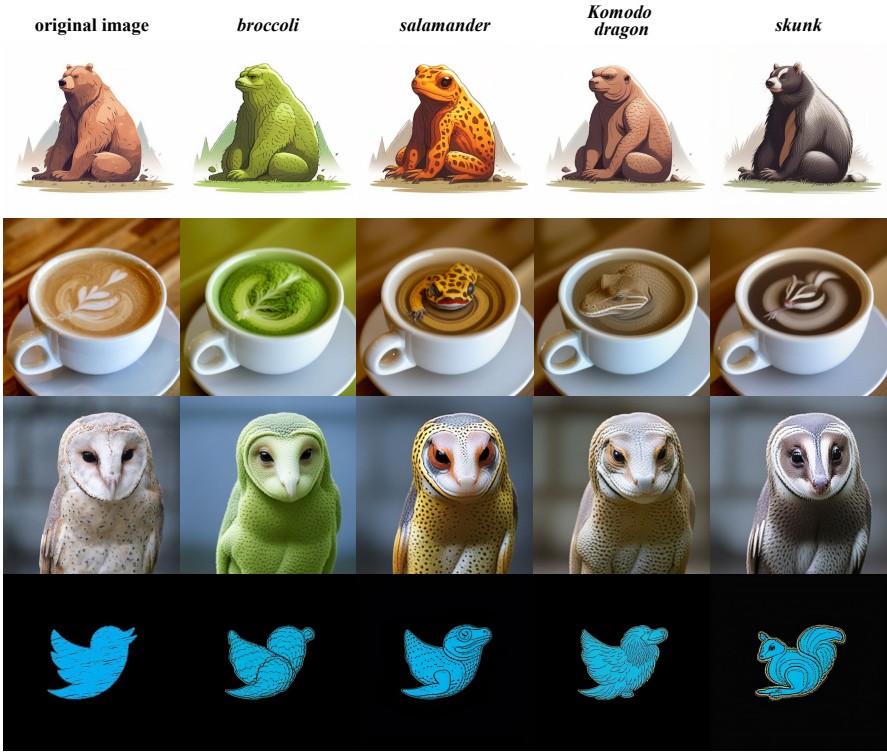

Figure 21: More visual Results.

## I   More Comparisons

In this section, we present additional results from our model and compare its performance against other methods.

In Fig. 24, we compare our results with those from the state-of-the-art T2I model DALL· E·3 assisted by Copilot. Our model shows superior performance when handling complex descriptive prompts for image editing. We observe that the competing model struggles to achieve results comparable to ours, particularly in maintaining the original structure and layout of images, despite adequate prompts.

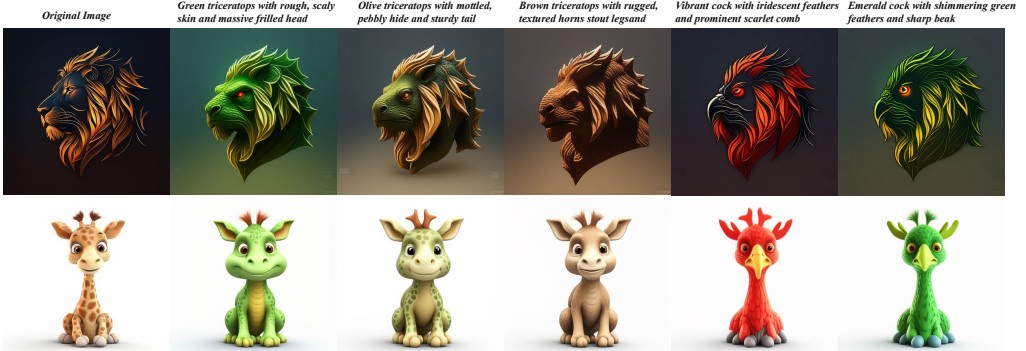

Figure 22: More visual results using complex prompt fusion.

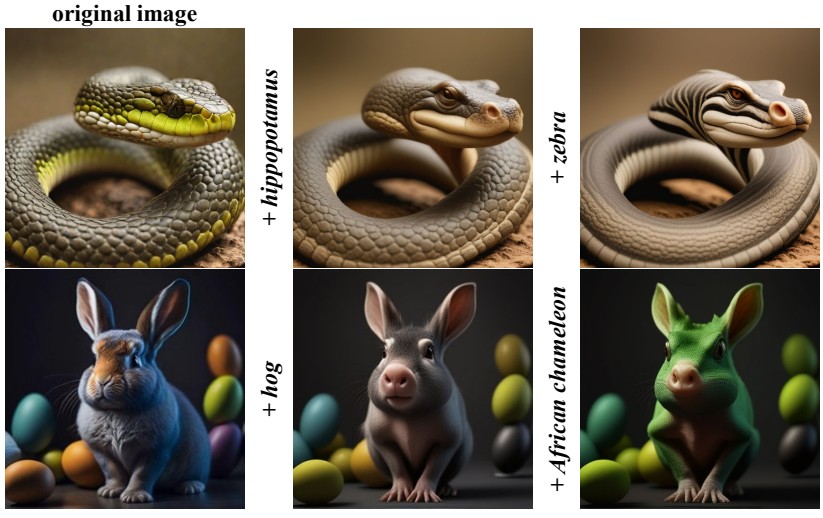

Figure 23: Fused results using three prompts.

In Figs. 25 and 26, we present additional comparison results with mixing methods. We observed that both MagicMix and ConceptLab tend to overly favor one category, as seen in examples like *Triceratops-Teddy Bear Toy* and *Anemone fish-Car*. Their generated images often lean more towards a single category.

Recently, subject-driven text-to-image generation focuses on creating highly customized images tailored to a target subject [18; 52; 9; 74]. These methods often address the task, such as multiple concept composition, style transfer and action editing [38; 8; 45]. In contrast, our approach aims to generate novel and surprising object images by combining object text with object images. Kosmos-G [45] utilize a single image input and a creative prompt to merge with specified text objects. The prompt is structured as " creatively fuse with object text," guiding the synthesis to innovatively blend image and text elements. Our findings indicate that Kosmos-G can sometimes struggle to maintain a balanced integration of original image features and text-driven attributes. In Fig. 27, the images generated by Kosmos-G often exhibit a disparity in feature integration.

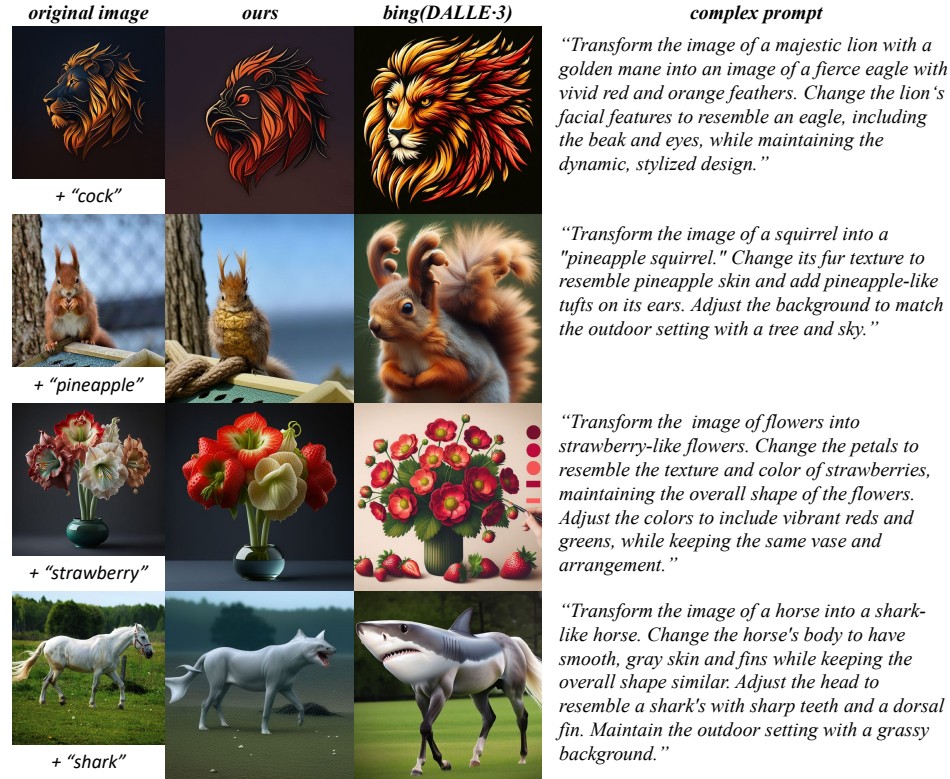

| original image | ours | bing(DALLE·3) | complex prompt |
|---|---|---|---|

+ "cock"

*"Transform the image of a majestic lion with a golden mane into an image of a fierce eagle with vivid red and orange feathers. Change the lion's facial features to resemble an eagle, including the beak and eyes, while maintaining the dynamic, stylized design."*

+ "pineapple"

*"Transform the image of a squirrel into a "pineapple squirrel." Change its fur texture to resemble pineapple skin and add pineapple-like tufts on its ears. Adjust the background to match the outdoor setting with a tree and sky."*

+ "strawberry"

*"Transform the image of flowers into strawberry-like flowers. Change the petals to resemble the texture and color of strawberries, maintaining the overall shape of the flowers. Adjust the colors to include vibrant reds and greens, while keeping the same vase and arrangement."*

+ "shark"

*"Transform the image of a horse into a shark-like horse. Change the horse's body to have smooth, gray skin and fins while keeping the overall shape similar. Adjust the head to resemble a shark's with sharp teeth and a dorsal fin. Maintain the outdoor setting with a grassy background."*

Figure 24: Comparisons with complex prompt editing.

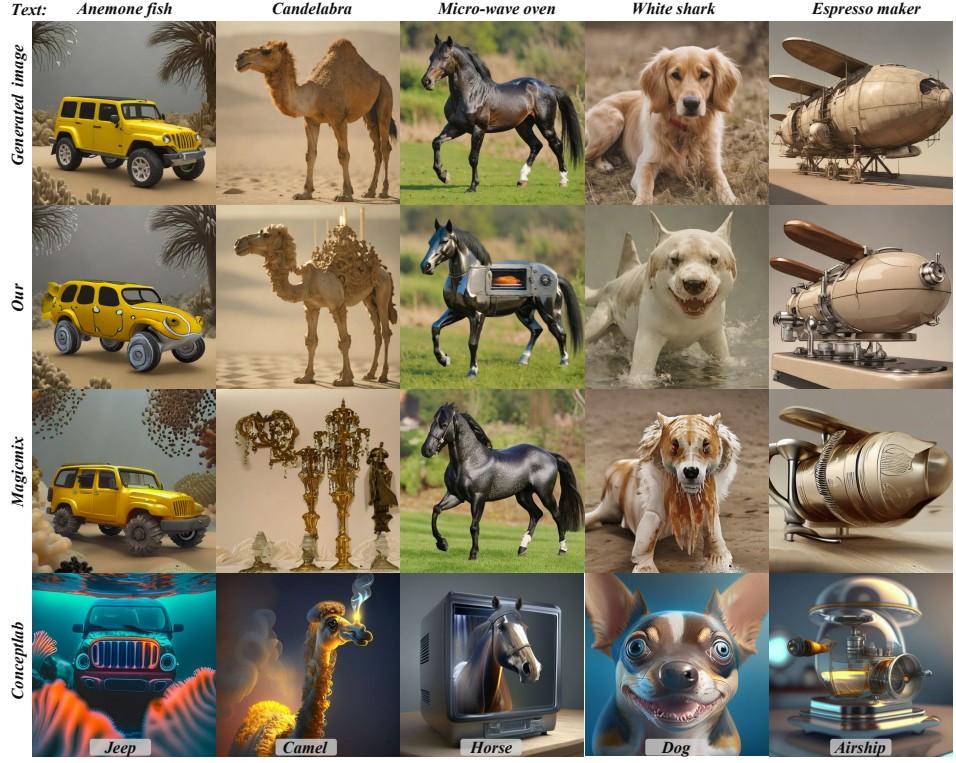

Figure 25: Comparison results of mixing methods using text-generated images.

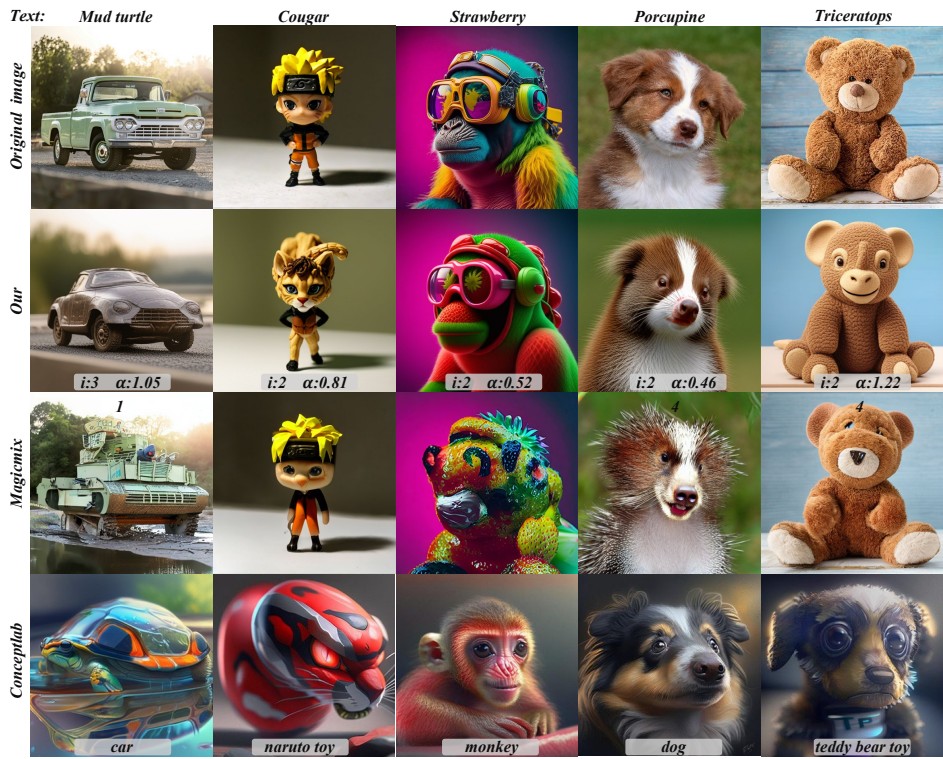

Figure 26: Further comparisons with mixing methods.

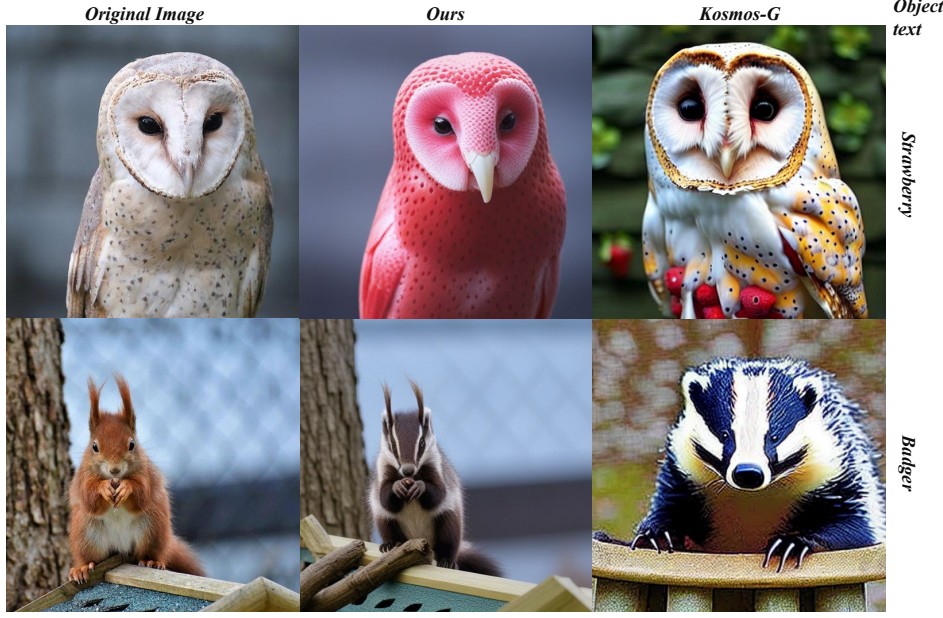

Figure 27: Comparisons with Subject-driven method.

