# OpenReview forum: "Novel Object Synthesis via Adaptive Text-Image Harmony"
_NeurIPS.cc/2024/Conference — NeurIPS 2024 poster_

### Official Review · Reviewer_VGMU · 2024-07-12

**Soundness:** 3
**Presentation:** 3
**Contribution:** 3
**Rating:** 6
**Confidence:** 3

**Summary:**

The paper works on novel object synthesis, namely, given an image and text prompt, the proposed method can generate a new image that contains the visual features from the given image and the textual information from the given prompt. The paper proposed a method names Adaptive Text-Image Harmony to tackle the task. The authors provide both qualitative and quantitate comparisons to show the effectiveness of the method.

**Strengths:**

The paper is easy to follow.

The author proposed an effective method that solves the challenging problem of novel object synthesis.

The paper provides extensive experiments, along with human evaluation, to justify the effectiveness of the approach.

Authors also provide detailed ablation for the design of the approach.

**Weaknesses:**

For the constructed dataset with 1,800 text-image pairs, will the authors release it? Could this dataset be used to fine-tune other baseline approaches to improve performance? If so, has this experiments been done?

For the baseline methods, e.g., InstructPix2Pix, did the authors directly use the released model or train a new one using the same text-to-image model as this work, which is SDXL-Turbo?

**Questions:**

Please refer to the weakness section.

**Limitations:**

Yes.

---

> ### Author Rebuttal · Authors · 2024-08-07
>
> **Q1: Dataset and fine-tune baselines.**
>
> **A1:** Thank you for your interest in our dataset and its potential applications.
>
> **Dataset Release:**
> We plan to release a dataset of 1,800 text-image pairs to the research community following acceptance. These pairs are created using an outer product on 60 different object texts and 30 different object images. Specifically, we selected 60 texts from the ImageNet dataset \[43\], detailed in Table 4, including categories such as kit fox, peacock, African chameleon, white shark, acorn, zucchini, and fire engine. The 30 images were chosen from the PIE-bench dataset \[25\], shown in Fig. 10, with each image corresponding to similarly categorized texts outlined in Table 5, such as Corgi, Duck, ladybug, flower vase, apple, jar, man, and Twitter logo. More details are provided in Appendix C. It should be noted that there are no ground truths for novel and surprising objects in this dataset created from text-image pairs.
>
> **No fine-tuning for all baselines and our ATIH method:**
> Due to the absence of ground truths, this dataset cannot be used to train any baselines or our ATIH method. Instead, it is solely for evaluating performance using four key popular metrics: CLIP text-image similarity (CLIP-T) \[39\], Dinov2 image similarity (Dino-I) \[36\], aesthetic score (AES) \[46\] and human preference score (HPS) \[56\], and our two proposed metrics: Fscore and balance similarities (Bsim). Our ATIH method can generate novel and surprising objects by designing an adaptive balance between the provided object text and object image.
>
> **Q2: Using released or SDXL-Turbo-trained models.**
>
> **A2:** Thanks. We directly used the released models for the baseline methods, including InstructPix2Pix, due to the following reasons. First, our dataset lacks ground truths for novel and surprising objects created from text-image pairs, preventing us from training or fine-tuning these baseline models. Second, using the officially released models allows us to provide a direct and unbiased comparison, as these models have been trained and optimized by their developers. Third, we employed the officially released SDXL-Turbo \[45\] as a base model to create novel and surprising objects using our ATIH method, since no ground truths were used to train the SDXL-Turbo model.
>
> Additionally, we manually adjusted the image and text scales of InstructPix2Pix, as shown in Figure R3. Our ATIH method demonstrates significantly better visualization compared to InstructPix2Pix, which displays objects with unnatural combinations.

---

> > ### Author Response · Authors · 2024-08-12
> >
> > ## Dear Reviewer VGMU
> > Thank you for taking the time to review our submission and providing us with constructive comments and a favorable recommendation. We would like to know if our responses adequately addressed your earlier concerns.
> >
> > Additionally, if you have any further concerns or suggestions, please feel free to let us know. We eagerly await your response and look forward to hearing from you.
> >
> > Thank you for your valuable time and consideration!
> >
> > Best regards,
> >
> > The authors

---

### Official Review · Reviewer_EP4q · 2024-07-12

**Soundness:** 2
**Presentation:** 3
**Contribution:** 2
**Rating:** 5
**Confidence:** 3

**Summary:**

This paper aims to generate novel objects based on a reference image and a conditional text prompt (e.g., a bottle (image) with a penguin (text) outlook). To this end, a method called Adaptive Text-Image Harmony (ATIH) is proposed to better align the conditional image and text. Experiments show that ATIH yields better results compared to existing state-of-the-art methods.

**Strengths:**

- Overall, the paper is easy to follow.
- Experiments in the main paper and supplementary materials try to be as thorough as possible to cover the most promising frameworks (e.g., ConceptLab) and ablation studies.
- On a high-level idea of fine-grained controllable image generation, I think this topic will attract interest in the image editing/image manipulation community.
- While the idea of manipulating attention layers of diffusion models is not new, I think this paper finds a way to (1) preserve the structure of the reference image as much as possible while (2) editing the image toward the given text -- which I consider a main contribution of this paper.

**Weaknesses:**

- As far as I understand, the image for text information in Figure 2 (e.g., "iron", "white shark") is for visualization purposes only. Actually, I think this might mislead readers (e.g., models can take two reference images as input).
- I think one thing the authors must consider is adjusting the strengths of existing baselines (e.g., InstructPix2Pix). If baselines allow for controlling the strength of the edit, it would be fairer if we had comparisons to that (e.g., similar to Figure 16, but for baselines).
- While in the main paper, the Golden Section Search algorithm seems to play an important role, it's unclear how important this algorithm is (e.g., can we simply replace this with simple average values). In other words, I think the current paper lacks an ablation study for this module.

**Questions:**

It'd be interesting if the authors can discuss how to make a better evaluation for this task. Overall, I think this task is ambiguous and difficult to evaluate (e.g., if you want to turn a cat into a peacock, is it better to (1) completely replace the cat with the peacock, or (2) yield a combination of the cat and the peacock?) How can we define which is a 'better alignment' with the user's intention? (This is an open-ended question and will not affect my recommendation for this paper).

**Limitations:**

The authors did briefly discuss the limitations in the Supplementary.

At this moment, I'd recommend this paper for acceptance, as I believe the task introduced by this paper outweighs the limitations in experiments and ablation studies.

---

> ### Author Rebuttal · Authors · 2024-08-07
>
> **Q1: Remove the misleading images.**
>
> **A1:** Thank you for your feedback regarding the potential misunderstanding in Figure 2. We acknowledge that the image for text information was intended for visualization purposes only, and we agree that it could mislead readers. Therefore, we will remove the images and present only the text in the same location.
>
> **Q2: Adjusting baseline strengths of InstructPix2Pix.**
>
> **A2:** Thank you for your suggestion. We agree that adjusting the strengths of existing baselines, such as InstructPix2Pix, is essential. Using 'cock' (text) and 'rabbit' (image), we manually adjusted the image strength (e.g., 1.0, 1.5, 2.0) and text strength (e.g., 1.5, 2.0, 2.5, 4.0, 4.5, 5.0, 6.5, 7.0, 7.5) for InstructPix2Pix, and the results are shown in Figure R3. Using the optimal image and text strengths of 1.5 and 5.0, respectively, InstructPix2Pix generates unnatural combinations, such as the head of the cock replacing the ears of the rabbit, as seen on the left of Figure R3. In contrast, our method produces natural and novel combinations, with the head and feet of the rabbit fused correspondingly with the head and feet of the cock, in the right of Figure R3.
>
> **Q3: Ablation study of Golden Section Search.**
>
> **A3:** Thank you for your insightful feedback regarding the role of the Golden Section Search algorithm.
>
> **Role of Golden Section Search:**
> In our approach, we designed a score function to evaluate the quality of the text-image fusion. We utilize the Golden Section Search algorithm to efficiently find the optimal parameters that maximize our score function, allowing us to achieve a well-balanced and harmonious fusion of text and image features.
>
> **Alternative Search Methods:**
> While the Golden Section Search is our chosen method for parameter optimization due to its efficiency and simplicity, we acknowledge that other methods, such as Ternary search \[ref-1\] or even Random search \[ref-2\], could potentially achieve similar results. The primary advantage of the Golden Section Search is its ability to converge to an optimal solution with relatively few iterations. In our task, it reaches convergence two iterations faster than the Ternary Search.
>
> **Importance and Flexibility:**
> The choice of search algorithm is flexible, and the primary contribution is the score function itself, which guides the search for optimal fusion parameters. The Golden Section Search provides a systematic approach, but its role can be substituted by other search methods if desired.
>
> **Ablation Study:**
> We conducted an ablation study comparing the Golden Section Search and Ternary Search, as shown in Figure R7. The images produced using the Golden Section Search exhibit a more novel fusion effect.
>
> \[ref-1\] Kiefer, J. (1953). Sequential Minimax Search for a Maximum. Proceedings of the American Mathematical Society, 4(3), 502-506.
>
> \[ref-2\] Solis, F. J., & Wets, R. J. B. (1981). Minimization by Random Search Techniques. Mathematics of Operations Research, 6(1), 19-30.
>
> **Q4: Discussing evaluation criteria for ambiguous tasks.**
>
> **A4:** Thank you for your understanding. We acknowledge that evaluating (1) complete replacement and (2) harmonious combination is challenging. For complete replacement, we can consider two metrics: the difference between the given object text and its generated object patch, and the harmony and naturalness of the entire image replaced by the given object text. To measure the difference, we can use CLIP text-image similarity (CLIP-T). For evaluating harmony and naturalness, we can employ the aesthetic score (AES) and the human preference score (HPS).
>
> For harmonious combination, our goal is to generate an object that incorporates characteristics of both A and B. If the generated object is unbiased towards either A or B, it indicates a balance between them. To achieve this, we first use CLIP text-image similarity (CLIP-T) and Dinov2 image similarity (Dino-I) to ensure the generated object has high similarities with both A and B. We then introduce a balance similarity metric (\(B\)sim) to measure the equilibrium between A and B. If the generated object shows a bias towards either, we adjust it using a scale factor \(k\).

---

> > ### Author Response · Authors · 2024-08-12
> >
> > ## Dear Reviewer EP4q
> > Thank you for taking the time to review our submission and providing us with constructive comments and a favorable recommendation. We would like to know if our responses adequately addressed your earlier concerns.
> >
> > Additionally, if you have any further concerns or suggestions, please feel free to let us know. We eagerly await your response and look forward to hearing from you.
> >
> > Thank you for your valuable time and consideration!
> >
> > Best regards,
> >
> > The authors

---

> > > ### Comment · Reviewer_EP4q · 2024-08-13
> > >
> > > Thanks authors for providing a detailed rebuttal.
> > >
> > > I have read the rebuttal and other reviews. I have decided to keep my positive rating for this paper as "Borderline Accept," as before.
> > >
> > > My justification for not giving a higher score is the lack of broader/thorough evaluation (as mentioned by other reviewers, which I agree). However, I find the task novel and interesting, so I am inclined to vote for acceptance.
> > >
> > > Please incorporate the discussion/evaluation into the main paper. Good luck!

---

> > > > ### Author Response · Authors · 2024-08-14
> > > >
> > > > Thank you for maintaining your positive rating. We sincerely appreciate your recognition of our work and your constructive feedback. We will incorporate the broader evaluation and discussion into the main paper as recommended.

---

### Official Review · Reviewer_yinU · 2024-07-12

**Soundness:** 3
**Presentation:** 2
**Contribution:** 2
**Rating:** 5
**Confidence:** 4

**Summary:**

The paper proposes a new method for harmonized image generation conditioned on both text and image conditions, leading to better performance than baselines mentioned in the paper.

**Strengths:**

The idea of using scale factor to combine different conditions is straightforward and reasonable. It is also

The qualitative results shows better image fidelity compared to some previous image editing methods.

**Weaknesses:**

The optimal combination between image and text condition is subjective. In the paper, all the generate examples have similar depth/layout/structure with input image condition, while with modified texture/appearance guided by the text. Assume the user want to generate "an image of a shark" with color/pattern similar to a given dog image, i.e. which is different from the example in Figure 2. It is not clear whether the proposed method can achieve that.

Furthermore,  some important baselines are missing:

1. The depth/structure information from the input image can be extracted and injected into the generation process with a ControlNet-like model [1]. This baseline is important because there is little structural change shown in the results of proposed  methods, which is the scenario that the well-known ControlNet is good at;

2. A naive way to generate with two conditions is using revised classifier-free guidance as [2]. The guidance strength from image and text can be controlled by hyper-parameters.

3. Another naive baseline is to only use image guidance before timestep t , and use text guidance after t. Results with different t will be interesting.


Subject-driven image generation and editing methods [3,4,5,6,7,8, 9] need to be discussed or at least mentioned, as these methods also aim to generate image guided by the reference images and text prompts.


[1]. Adding Conditional Control to Text-to-Image Diffusion Models. Lvmin Zhang, Anyi Rao, Maneesh Agrawala.

[2]. InstructPix2Pix: Learning to Follow Image Editing Instructions. Tim Brooks, Aleksander Holynski, Alexei A. Efros.

[3]. An Image is Worth One Word: Personalizing Text-to-Image Generation using Textual Inversion. Rinon Gal, Yuval Alaluf, Yuval Atzmon, Or Patashnik, Amit H. Bermano, Gal Chechik, Daniel Cohen-Or.

[4]. DreamBooth: Fine Tuning Text-to-Image Diffusion Models for Subject-Driven Generation. Nataniel Ruiz Yuanzhen Li Varun Jampani Yael Pritch Michael Rubinstein Kfir Aberman.

[5]. Subject-driven Text-to-Image Generation via Apprenticeship Learning. Wenhu Chen, Hexiang Hu, Yandong Li, Nataniel Ruiz, Xuhui Jia, Ming-Wei Chang, William W. Cohen.

[6]. Customization Assistant for Text-to-image Generation. Yufan Zhou, Ruiyi Zhang, Jiuxiang Gu, Tong Sun.

[7]. Subject-Diffusion:Open Domain Personalized Text-to-Image Generation without Test-time Fine-tuning. Jian Ma, Junhao Liang, Chen Chen, Haonan Lu.

[8]. Multi-Concept Customization of Text-to-Image Diffusion. Nupur Kumari, Bingliang Zhang, Richard Zhang, Eli Shechtman, Jun-Yan Zhu.

[9]. Kosmos-G: Generating Images in Context with Multimodal Large Language Models. Xichen Pan, Li Dong, Shaohan Huang, Zhiliang Peng, Wenhu Chen, Furu Wei.

**Questions:**

Can the proposed method generate object whose shape is guided by the text prompt, while appearance/texture is guided by a reference image?

**Limitations:**

Yes

---

> ### Author Rebuttal · Authors · 2024-08-07
>
> **Q1: Results by the different assumption.**
>
> **A1:** Thank you for your suggestion. Inspired by \[ref-1\], we only inject the late self-attention layers in our method while keeping other settings the same. This simple adjustment enables our model to effectively modify the visual style of generated images while maintaining the desired structural features. In Figure R3, our method can create an image where the shape is guided by the text ("shark"), and the appearance (color/pattern) is influenced by the referenced image of a dog.
>
> There are some key differences between our method and Visual Style Prompting (VSP) \[ref-1\]. First, our method introduces a new task that combines an object text with an object image to create a new object image, while VSP is a novel approach that guides the desired style using a reference image. Second, we propose ATIH to achieve a creative and harmonious fusion of two distinct objects, resulting in surprising and imaginative combinations. VSP aims to produce a diverse range of images while maintaining specific style elements and nuances. Third, while VSP primarily adjusts the visual style, our results demonstrate the novel combination of object text and object image in Figure R3.
>
> \[ref-1\] Visual Style Prompting with Swapping Self-Attention. arXiv preprint arXiv:2402.12974.
>
> **Q2: Results by using ControlNet-like model to inject structural information.**
>
> **A2:** Thank you for the opportunity to clarify the performance of our method compared to ControlNet, particularly in terms of achieving a balanced and semantically coherent image synthesis.
>
> **Comparison with ControlNet:**
> We rigorously compared our method with ControlNet to evaluate their capabilities in handling complex text-image fusion tasks in Figure R2. Our findings indicate significant differences in how both approaches manage the fusion process. ControlNet tends to maintain the structure from depth or edge maps well but struggles with semantic integration, especially when faced with complex prompts, failing to achieve a seamless blend. By contrast, our method utilizes the full spectrum of RGB image features, including color and texture, alongside structural data.
>
> **Q3: Results by controlling image and text strengths.**
>
> **A3:** Thank you for highlighting the relevance of adjusting the guidance strength in image and text synthesis, as seen in methods like InstructPix2Pix.
>
> **Comparison with InstructPix2Pix:**
> We manually adjusted the image and text strengths for InstructPix2Pix to match the controls of our method. We varied the image strength (e.g., 1.0, 1.5, 2.0) and text strength from 1.5 to 7.5 and observed the outcomes. In Figure R3, at optimal settings of image strength 1.5 and text strength 5.0, InstructPix2Pix produced the best fusion, but it is unnatural, such as replacing the rabbit's ears with the rooster's head. In contrast, our results are new and natural combinations of the rabbit and the rooster, demonstrating superior visual synthesis compared to InstructPix2Pix.
>
> More importantly, our method can automatically iterate to the optimal fusion of image and text, eliminating the need for manual parameter adjustments. This automation significantly enhances the usability and efficiency of our synthesis process, allowing for seamless generation of harmonious images that meet both aesthetic and semantic criteria.
>
> **Q4: Baseline using only image guidance before t, text guidance after t.**
>
> **A4:** Thank you for suggesting the exploration of different guidance strategies.
>
> **Results and Analysis:**
> Figure R6 illustrates the transition and effects of this guidance strategy: At \(t=0\), the output is purely influenced by the text, which in this case describes an ostrich. Starting at \(t=1\), image guidance is introduced, leading to a more complex synthesis process where the features of the deer image begin to merge with the previously established text-based ostrich features.
>
> Figure R6 shows the results that the integration is not fully balanced, as the distinctive features of the ostrich—such as its neck and feather texture—are not as prominent as desired. This indicates a need for more refined control over the balance and timing of text and image guidance.
>
> **Q5: Discussing subject-driven generation methods.**
>
> **A5:** Thank you for mentioning the subject-driven methods. Recently, subject-driven text-to-image generation focuses on creating highly customized images tailored to a target subject \[3,4,5,6\]. These methods often address the task, such as multiple concept composition, style transfer, and action editing \[7,8,9\]. In contrast, our approach aims to generate novel and surprising object images by combining object text with object images.
>
> **Comparison with Kosmos-G:**
> Kosmos-G \[9\] utilizes a single image input and a creative prompt to merge with specified text objects. The prompt is structured as “\<i\> creatively fuse with {object text},” guiding the synthesis to innovatively blend image and text elements. Our findings indicate that Kosmos-G can sometimes struggle to maintain a balanced integration of original image features and text-driven attributes. In Figure R4, the images generated by Kosmos-G often exhibit a disparity in feature integration.
>
> **Q6: Can the method use text for shape and image for texture?**
>
> **A6:** Thanks. Altering the shape of a given object image is quite challenging. To the best of our knowledge, there is no method that performs this task well. Fortunately, our method can generate a new object with slight deformations, such as the axolotl's mouth in Figure 2. However, our method still cannot significantly deform the shape of the object image, as we incorporate all self-attention to preserve image information during the diffusion process. We will study this large deformation in the future work.

---

> > ### Author Response · Authors · 2024-08-12
> >
> > ## Dear Reviewer yinU
> > Thank you for taking the time to review our submission and providing us with constructive comments and a favorable recommendation. We would like to know if our responses adequately addressed your earlier concerns.
> >
> > Additionally, if you have any further concerns or suggestions, please feel free to let us know. We eagerly await your response and look forward to hearing from you.
> >
> > Thank you for your valuable time and consideration!
> >
> > Best regards,
> >
> > The authors

---

> > > ### Comment · Reviewer_yinU · 2024-08-13
> > >
> > > I have read the rebuttal and would like to thank the authors for the additional results, especially those on ControlNet and editing methods, which addressed some of my concerns. I have adjusted the score accordingly.

---

> > > > ### Author Response · Authors · 2024-08-13
> > > >
> > > > Thank you for raising the score. We sincerely appreciate your recognition of our work and effort.

---

### Official Review · Reviewer_xaqc · 2024-07-13

**Soundness:** 3
**Presentation:** 3
**Contribution:** 3
**Rating:** 5
**Confidence:** 4

**Summary:**

The paper introduces an innovative method to generate new object images by combining textual descriptions with corresponding images. Addressing the common imbalance in diffusion models between text and image inputs, the authors propose the Adaptive Text-Image Harmony (ATIH) method. This method balances text and image features during cross-attention, ensuring a harmonious integration.
Key to this method are the scale factor (α) and injection step (i), which adjust the influence of text features and preserve image information. A novel similarity score function maximizes and balances the similarities between the generated image and the input text/image, while a balanced loss function with a noise parameter optimizes the trade-off between editability and fidelity of the synthesized image.
Validated through extensive experiments on datasets like PIE-bench and ImageNet, ATIH outperforms state-of-the-art techniques in creative object synthesis. The method showcases remarkable results, such as creating a dog-lobster or a rooster with an iron texture, demonstrating its innovative and high-quality image generation capabilities. The framework includes a Text-Image Diffusion Model (TIDM) using a pre-trained SDXL Turbo model with dual denoising branches. By treating sampling noise as a learnable parameter and designing a balance loss function, the method enhances image fidelity and editability. The ATIH method adaptively adjusts the scale factor and injection step to balance text and image similarities, using a similarity score function and the Golden Section Search algorithm to find optimal parameters.

**Strengths:**

1. Originality: The paper introduces a novel method for combining textual descriptions with corresponding images to generate new object images. The originality lies in the Adaptive Text-Image Harmony (ATIH) method, which effectively balances text and image features during the cross-attention mechanism in diffusion models. This innovative approach addresses a significant challenge in existing methods— the imbalance between text and image inputs— and offers a robust solution to achieve harmonious integration. The introduction of a scale factor (α) and an injection step (i) to fine-tune this balance further highlights the creativity and novelty of the approach.

2. Quality: The paper demonstrates high quality through comprehensive experiments and rigorous validation. The authors provide detailed methodological explanations and present clear experimental results on datasets such as PIE-bench and ImageNet. The inclusion of a balanced loss function with a noise parameter to optimize the trade-off between editability and fidelity showcases the thoroughness of the approach. The comparison with state-of-the-art techniques further establishes the effectiveness and superiority of the ATIH method. The experimental results, including examples like a dog-lobster and a rooster with iron texture, illustrate the method's capability to produce high-quality, innovative, and coherent images.

3. Clarity: The paper is well-written and clearly structured. The authors provide a concise and comprehensive introduction to the problem, followed by a detailed explanation of the proposed ATIH method. The use of figures and diagrams, such as the framework of the Text-Image Diffusion Model (TIDM) and the results of the experiments, aids in understanding the methodology and its impact. The step-by-step breakdown of the methodology, along with the clear presentation of experimental results, ensures that readers can easily follow the proposed approach and its benefits. The balanced loss function and similarity score function are well-explained, contributing to the overall clarity of the paper.

4. Significance: The proposed method's capability to outperform existing techniques and produce high-quality, innovative results underscores its importance and impact on the broader field of text-to-image synthesis.

**Weaknesses:**

1.  While the paper discusses a novel algorithm for text-image harmony, the experimental validation focuses heavily on visual outcomes without substantial quantitative backing or comparisons against baselines using metrics relevant to the text-image synthesis field. Including more diverse and robust statistical evaluations could strengthen the claims about the algorithm's effectiveness.

2. The experiments primarily utilize a specific set of images and texts, which might not adequately represent the variety of real-world scenarios where such an algorithm could be applied. Expanding the dataset to include a wider range of text and image pairs, especially challenging ones, would help in understanding the algorithm's limitations and strengths better.

3. The paper could benefit from a deeper exploration of the robustness of the proposed method, particularly in how it handles edge cases or unusual text-image pairs. This includes testing the method's performance on non-ideal or adversarial inputs to gauge its resilience and adaptability.

4. More detailed comparative analysis with existing methods, especially recent advancements in text-to-image synthesis and image editing technologies, would provide a clearer picture of where the proposed method stands in terms of innovation and improvement. Specific examples of where it outperforms and underperforms can guide future development.

5. The methodology section could be expanded to include more technical details about the implementation, which would aid in reproducibility. For instance, details on parameter settings, algorithmic steps that were particularly effective, and potential pitfalls in the implementation could be highlighted.

6. The paper mentions user studies but does not delve deeply into how the feedback from these studies was integrated into the algorithm refinement. Elaborating on this process could provide insights into the user-centric development of the algorithm and its practical usability.

**Questions:**

1. Please include a detailed description of the selection criteria for the text-image pairs used in your experiments. It is important to demonstrate the diversity and representativeness of these pairs to ensure the robustness of your Adaptive Text-Image Harmony (ATIH) method.

2. It would be beneficial to expand the comparative analysis section with more detailed results, including statistical significance tests. This would provide a clearer picture of how your method improves over existing methods, underlining its novelty and effectiveness.

3. Please discuss the performance of your method under non-ideal conditions, such as complex or abstract text descriptions. Identifying and describing limitations observed in such scenarios could guide future improvements and research.

4. Including comprehensive implementation details, particularly regarding parameter settings and platform-specific optimizations, would significantly aid in the reproducibility and independent verification of your results.

5. Identifying specific edge cases and failure modes encountered during testing and development would help in understanding the practical limitations of your model. This information is crucial for setting realistic expectations and guiding future research directions.

**Limitations:**

The limitations have not been addressed.

---

> ### Author Rebuttal · Authors · 2024-08-07
>
> **Q1: More quantitative validation**
>
> **A1:** Thank you for your thoughtful feedback. We recognize the importance of quantitative metrics in validating our algorithm's effectiveness.
>
> We use four widely-used metrics: CLIP-T \[39\], Dino-I \[36\], AES \[46\], and HPS \[56\]. Additionally, we propose two new metrics: Fscore and balance similarity (Bsim), to evaluate our ATIH method.
>
> CLIP-T and Dino-I measure similarity between the generated image and the object text or image, indicating how well the generated object incorporates characteristics of both sources. The balance similarity metric (Bsim) assesses the equilibrium between text and image fusion. The Fscore considers both higher CLIP-T and Dino-I, and the best balance. AES and HPS evaluate the aesthetic quality and human preference for the generated object.
>
> Table 1 shows that we achieve relatively high CLIP-T and Dino-I scores, though not the highest single scores. However, we obtain the best Bsim, Fscore, AES, and HPS. We also conducted a comprehensive user study to gather qualitative feedback on the perceived quality and creativity of the synthesized images.
>
> **Q2: Expanding dataset.**
>
> **A2:** Thank you for your valuable feedback regarding the dataset. We agree that testing the algorithm across a broader range of scenarios can provide more insights. Our dataset is created using an outer product on 60 different object texts and 30 different object images, representing widely-used categories. For example, the porcupine-bottle in Figure 1 illustrates the diverse text-image pairs.
>
> We selected 30 images from the PIE-bench dataset, representing a diverse set of distinct categories, each with a clear and identifiable subject in Figure 10 and Table 5, including categories such as Corgi, jar, and man.
>
> For the text categories, we utilized ImageNet and employed ChatGPT to filter and select 40 distinct animal categories and 20 non-animal categories, in Table 4. These include categories such as kit fox, peacock, acorn, and zucchini. This selection was intended to cover a broad range of typical scenarios while ensuring that the algorithm could effectively handle both common and varied text-image combinations.
>
> **Q3: Handling edge cases and unusual text-image pairs**
>
> **A3:** Thank you for your insightful comments on our method's performance with edge cases and unusual text-image pairs.
>
> To address edge cases, our method effectively captures prominent features like colors and basic object forms from complex textual descriptions. Figure R1 shows a well-defined edge structure for the fawn image and the text 'Green triceratops with rough, scaly skin and massive frilled head.'
>
> Handling unusual text-image pairs is challenging. Our method relies on semantic correlations within the diffusion feature space. When the semantic match is weak, it tends to produce mere texture changes rather than deeper transformations. This suggests our approach may struggle with transformations between categories with weak semantic associations. Future work could enhance semantic matching between categories to improve generalizability. See Appendix B.
>
> **Q4: More detailed comparative analysis with existing methods.**
>
> **A4:** Thank you for your insightful feedback. The core of our method is an adaptive text-image harmony approach that balances the similarities between the generated object and the given text-image pair. Unlike existing methods like MasaCtrl, InfEdit, ConceptLab, and MagicMix, our approach ensures better fusion, especially in challenging cases such as the African chameleon-bird, minimizing distortions and maintaining high image quality.
>
> Even with manual adjustments to match our controls, InstructPix2Pix produced unnatural results. In Figure R3, even at optimal settings (image strength 1.5, text strength 5.0), InstructPix2Pix's fusion was still unnatural, such as replacing the rabbit's ears with the rooster's head.
>
> **Q5: Technical details of our method.**
>
> **A5:** Thank you for your suggestion.The technical details of our method are provided in Section 3 and Subsection 4.1, and summarized in Algorithm 1 in Appendix F.
>
> We implemented our ATIH method using the SDXLturbo, achieving a processing time of ten seconds per sample. The input consists of a subject-specific object image and a fused object text.
>
> Key parameters in our experiments include:
> - The balance parameter λ is set to 125.
> - The scale factor k is set to 2.3.
> - The similarity thresholds $I _{sim}^{min}$ and  $I _{sim}^{max}$ are set to 0.45 and 0.85, respectively.
>
> **Q6: User feedback integrated into algorithm refinement.**
>
> **A6:** Thank you for emphasizing the importance of user studies in our research. We clarify that user studies are used solely to evaluate the performance of our ATIH method, not to refine it.
> User studies aim to evaluate and validate the performance of our ATIH method. These studies were designed to gather qualitative feedback on the novelty, coherence, and visual appeal of the generated images, rather than for algorithm refinement. We plan to incorporate it into subsequent versions to enhance usability and customization.
>
> **Q7: Statistical significance tests.**
>
> **A7:** Thank you for your suggestion to provide a more detailed comparative analysis. We conducted H tests to determine the statistical significance of performance differences between our method and other methods across various metrics. The results are presented in Table R1. For example, AES and HPS: Against Instructpix2pix, our method demonstrates statistically significant differences with H statistics of 268.57$(p < 10^{-59})$ for AES and 39.63 $(p < 10^{-9})$ for HPS, indicating potential improvements in aesthetic quality and human preference scoring. These results validate the effectiveness of our approach, demonstrating its superiority across multiple critical metrics.

---

> > ### Author Response · Authors · 2024-08-12
> >
> > ## Dear Reviewer xaqc
> > Thank you for taking the time to review our submission and providing us with constructive comments and a favorable recommendation. We would like to know if our responses adequately addressed your earlier concerns.
> >
> > Additionally, if you have any further concerns or suggestions, please feel free to let us know. We eagerly await your response and look forward to hearing from you.
> >
> > Thank you for your valuable time and consideration!
> >
> > Best regards,
> >
> > The authors

---

> > ### Comment · Reviewer_xaqc · 2024-08-13
> > **Response for authors**
> >
> > Thanks, I've reviewed the author's response. However, I still have several concerns regarding this paper. The primary contributions—a scale factor to balance text and image features in cross-attention and a novel similarity score function—while further clarified with specific parameter values, lack sufficient theoretical justification. Besides, the comparison of the proposed method with others, while claiming superiority in fusion quality, appears to rely on somewhat selective examples (e.g., African chameleon-bird). A broader range of comparative cases or more challenging scenarios would provide a more comprehensive understanding of the method's strengths and weaknesses.

---

> > > ### Author Response · Authors · 2024-08-13
> > >
> > > Thank you for your valuable time and consideration. We would like to clarify the theoretical justification and broader comparative examples.
> > >
> > > **Q1: Sufficient Theoretical Justification.**
> > >
> > > **A1:** Our theoretical framework is grounded in a balance theory applied to multimodal data, specifically text and image, for the purpose of generating novel objects. Given an object described by text, $O_T$, and an object represented by an image, $O_I$, our goal is to synthesize a new object, $O(\alpha,i)$, parameterized by $\alpha$ and $i$. This synthesis aims to balance the distances (or similarities) between the new object and the original modalities. Ideally, the distance between $O(\alpha,i)$ and $O_T$ should be equal to the distance between $O(\alpha,i)$ and $O_I$, that is,
> > >
> > > $$
> > > d(O(\alpha,i),O_T)=d(O(\alpha,i),O_I),
> > > $$
> > >
> > > where $d(\cdot,\cdot)$ represents the similarity distance between text/image and image. For simplicity, we denote the similarity distance between the image $O_I$ and the created image $O(\alpha,i)$ as $I_{\text{sim}}(\alpha,i) = d(O_I, O(\alpha,i))$, and the similarity between the text $O_T$ and the created image $O(\alpha,i)$ as $T_{\text{sim}}(\alpha,i) = d(O_T, O(\alpha,i))$.
> > >
> > > In practice, to address the inconsistencies between text and image modalities, we introduce a scaling factor, $k$, which mitigates these discrepancies and defines a balance function,
> > >
> > > $$
> > > F_\text{balance}=|I_{\text{sim}}(\alpha,i)-k\cdot T_{\text{sim}}(\alpha,i)|,
> > > $$
> > >
> > > where $|\cdot|$ is an absolute value. When $F_\text{balance}$ is small, there is a greater balance between $O(\alpha,i)$ and $O_T/O_I$. Additionally, we aim for the generated novel object to increasingly incorporate the information from both $O_T$ and $O_I$. This implies that higher values of $I_{\text{sim}}(\alpha,i)$ and $k\cdot T_{\text{sim}}(\alpha,i)$ correspond to more comprehensive information being retained. Thus, we define an information function,
> > >
> > > $$
> > > F_\text{information}=I_{\text{sim}}(\alpha,i)+k\cdot T_{\text{sim}}(\alpha,i),
> > > $$
> > >
> > > By combining the equations for $F_\text{balance}$ and $F_\text{information}$, we define a score function
> > >
> > > $$
> > > F(\alpha,i) = F_\text{information} - F_\text{balance} = \underset{\text{maximize similarities}}{I_{\text{sim}}(\alpha,i) + k \cdot T_{\text{sim}}(\alpha,i)} - \underset{\text{balance similarities}}{|I_{\text{sim}}(\alpha,i) - k \cdot T_{\text{sim}}(\alpha,i)|}
> > > $$
> > >
> > >
> > > Finally, we use the score function $F(\alpha,i)$ to adaptively adjust the parameters, aiming to maximize the score, which corresponds to identifying the novel object $O$.
> > >
> > > To further substantiate our theoretical claims, extensive experiments were conducted to demonstrate the effectiveness of our scoring function. The quantitative results, as shown in Table 1, indicate that our approach outperforms others in terms of Aesthetic Score (AES), Human Preference Score (HPS), Fscore, and Balance Similarity ( Bsim). These outcomes highlight our method's excellence in enhancing the visual appeal and artistic quality of images, while also aligning more closely with human preferences and understanding in object fusion.
> > >
> > > **Q2: Broaden comparisons.**
> > >
> > > **A2:** Thank you for your mention. Here we explain the broader range of comparative examples as follows.
> > >
> > > **Broader Examples**: In our manuscript, supplementary materials, and rebuttal PDF, we have provided over 60 examples showcasing the versatility and robustness of our approach. These examples cover a wide array of categories, including *Cross-Species Fusions*, such as the combination of a corgi and a pineapple, and *Inanimate and Living Fusions*, like the fusion of a cup with a skunk.
> > >
> > > **Complex Text-Driven Fusions**: Figure R1 in our rebuttal provides additional examples of our method’s performance with complex text-driven fusions.
> > >
> > > **Demo and Code**: To facilitate further validation and exploration of our method's capabilities, we plan to release our code and a demo upon acceptance of the paper. This will enable academic and industry practitioners to replicate our results, ensuring transparency and reproducibility.
> > >
> > > **Weaknesses**: Our method relies on the semantic correlation between the original and transformed content within the diffusion feature space. When the semantic match between two categories is weak, our method tends to produce mere texture changes rather than deeper semantic transformations. This limitation suggests that our approach may struggle with transformations between categories with weak semantic associations. This limitation is detailed in Appendix B and is also mentioned on line 294.
> > >
> > > We thank you again for your feedback and hope that our explanation can make you better understand our contribution and efforts.
> > >
> > > Best regards,
> > >
> > > The authors

---

### Author Rebuttal · Authors · 2024-08-07

We thank all reviewers and chairs for their time, constructive comments, and recognition of our work. We sincerely hope that all reviewers can support our work, as this paper proposes a novel and reasonable method (**Reviewers xaqc and yinU**) to solve an interesting and challenging task (**Reviewers EP4q and VGMU**) and produces extensive, high-quality, innovative results (**Reviewers xaqc, yinU, VGMU**). Special thanks to \( **Reviewers EP4q**) for recognizing that the significance of the task introduced by this paper outweighs the limitations in experiments and ablation studies.

Our **Main Contribution** lies in the following three parts.

1. To the best of our knowledge, we are the first to propose an adaptive text-image harmony method for generating novel object synthesis. Our key idea is to achieve a balanced blend of object text and image by adaptively adjusting a scale factor and an injection step in the inversion diffusion process, ensuring effective harmony.

2. We introduce a novel similarity score function that incorporates the scale factor and injection step. This function aims to balance and maximize the similarities between the generated image and the input text/image, achieving a harmonious integration of text and image.

3. Our approach demonstrates superior performance in creative object combination compared to state-of-the-art image-editing and creative mixing methods. Examples of these creative objects include *sea lion-glass jar*, *African chameleon-bird*, and *mud turtle-car*.

---

### Decision · Program_Chairs · 2024-09-25

**Decision:**

Accept (poster)

**Comment:**

The paper introduces the Adaptive Text-Image Harmony (ATIH) method for generating novel object images by balancing text and image inputs in diffusion models. Reviewers appreciated the introduction of a scale factor and similarity score function but raised concerns about the lack of theoretical justification and the limited generalizability of the experimental results. In response, the authors provided additional quantitative metrics, expanded the dataset description, and compared their method with more baselines, such as ControlNet and InstructPix2Pix. They also committed to releasing the full dataset for future research. With generally positive ratings (5, 5, 5, 6), the paper is recommended for acceptance, pending the suggested revisions and dataset release, provided the authors make the suggested revisions and release the dataset as stated.